# CCL2 mobilizes ALIX to facilitate Gag-p6 mediated HIV-1 virion release

**David O Ajasin[1†], Vasudev R Rao[1†], Xuhong Wu[2], Santhamani Ramasamy[1], Mario Pujato[3], Arthur P Ruiz[1], Andras Fiser[4], Anne R Bresnick[3], Ganjam V Kalpana[2], Vinayaka R Prasad[1]\***

[1]Department of Microbiology and Immunology, Albert Einstein College of Medicine, Bronx, United States; [2]Department of Genetics, Albert Einstein College of Medicine, Bronx, United States; [3]Department of Biochemistry, Albert Einstein College of Medicine, Bronx, United States; [4]Department of Systems and Computational Biology, Albert Einstein College of Medicine, Bronx, United States

**Abstract** Cellular ESCRT machinery plays pivotal role in HIV-1 budding and release. Extracellular stimuli that modulate HIV-1 egress are currently unknown. We found that CCL2 induced by HIV-1 clade B (HIV-1B) infection of macrophages enhanced virus production, while CCL2 immuno-depletion reversed this effect. Additionally, HIV-1 clade C (HIV-1C) was refractory to CCL2 levels. We show that CCL2-mediated increase in virus production requires Gag late motif LYPX present in HIV-1B, but absent in HIV-1C, and ALIX protein that recruits ESCRT III complex. CCL2 immuno-depletion sequestered ALIX to F-actin structures, while CCL2 addition mobilized it to cytoplasm facilitating Gag-ALIX binding. The LYPX motif improves virus replication and its absence renders the virus less fit. Interestingly, novel variants of HIV-1C with PYRE/PYKE tetrapeptide insertions in Gag-p6 conferred ALIX binding, CCL2-responsiveness and enhanced virus replication. These results, for the first time, indicate that CCL2 mediates ALIX mobilization from F-actin and enhances HIV-1 release and fitness.

DOI: https://doi.org/10.7554/eLife.35546.001

**\*For correspondence:**
vinayaka.prasad@einstein.yu.edu

[†]These authors contributed equally to this work

**Competing interests:** The authors declare that no competing interests exist.

## Introduction

HIV-1 budding and release are mediated by the host ESCRT machinery, which facilitates membrane scission at the plasma membrane (*Votteler and Sundquist, 2013*). HIV-1 Gag polyprotein includes two motifs in the C-terminal p6 - late motif I (PTAP) and late motif II (LYPX) - which recruit host proteins TSG-101 (*Pornillos et al., 2002*; *VerPlank et al., 2001*) and ALIX (*Usami et al., 2007*; *Fisher et al., 2007*; *Munshi et al., 2007*), respectively. TSG101 is a subunit of the cellular ESCRT-I complex and ALIX is an accessory protein of ESCRT-III complex. These two proteins physically link late domains I and II of HIV-1 Gag-p6 to ESCRT I and ESCRT-III complexes, respectively (*Sun et al., 2015*) (*Strack et al., 2003*) and are both essential for membrane fission steps like MVB biogenesis, cytokinesis and virus budding. During cytokinesis, ESCRT I complex, once loaded, recruits the ESCRT II complex, which in turn recruits members of ESCRT III (CHMP6 and CHMP2/4) leading to the terminal events of membrane fission.

CCL2 is a potent β-chemokine induced upon HIV-1 infection of primary human macrophages (*Mengozzi et al., 1999*), as a host innate immune response designed to attract monocytes and T lymphocytes to trigger inflammation against the pathogen. CCL2 mRNA is elevated in both macrophages and PBMCs upon HIV-1 infection (*Mengozzi et al., 1999*; *Vázquez et al., 2005*; *Wetzel et al., 2002*; *Woelk et al., 2004*). Furthermore, plasma levels of CCL2 strongly correlate with plasma viral loads in HIV-infected individuals (*Weiss et al., 1997*). In a thorough analysis of a large number of host proinflammatory genes to identify markers that correlate with HIV-1 viral loads,

*CCL2* upregulation and elevated CCL2 protein levels were found to be specifically associated with viremic patients (*Ansari et al., 2006*). While these results suggest correlation of CCL2 to viremia, other results indicate a direct effect of CCL2 on viral replication. CCL2 addition enhances HIV-1 replication in cultured CD4$^+$ T cells (*Kinter et al., 1998*) and in macrophages (*Fantuzzi et al., 2003*), neutralization of CCL2 inhibits virus release and leads to intracellular accumulation of HIV-1 Gag (*Fantuzzi et al., 2003*). However, the mechanism by which CCL2 influences HIV-1 replication is unknown.

Here, we investigated the link between CCL2 and virus production. Furthermore, we compared the effects of CCL2 on clade C HIV-1 (HIV-1C) and HIV-1B side by side, as all the studies outlined above have been limited to HIV-1 clade B (HIV-1B). It is intriguing to note that infection of macrophages with clade C HIV-1 (HIV-1C), unlike that of HIV-1B, is not associated with a robust CCL2 induction (*Campbell et al., 2007*; *Rao et al., 2008*). However, it was unclear how this lack of CCL2 induction affects HIV-1 C replication. In this report, we show that addition of CCL2 resulted in mobilization of ALIX associated with F-actin to the cytoplasm making it available to bind HIV-1 Gag-p6, which consequently enhanced virion release. Furthermore, we found that immuno-depletion of CCL2 led to a dramatic colocalization of ALIX with F-actin structures, which was associated with decrease of virion release. On the contrary, we found that HIV-1C is refractory to CCL2 levels and that the inability to exploit the increased availability of ALIX is due to a lack of the LY dipeptide in late domain II or LYPX motif, in it's Gag-p6.

HIV-1C is known to be less fit than HIV-1B or other group M HIV-1 viruses (*Ariën et al., 2005*; *Ariën et al., 2007*). While the basis for this reduced fitness has been reported to be inefficient virus entry due to its *Env* gene (*Ball et al., 2003*), there is increasing evidence that a late event such as virus release may also be inefficient. For example, comparative study of a large number of HIV-1C and HIV-1B isolates has concluded that the absence of LY dipeptide is responsible for the reduced replication fitness of HIV-1C (*Kiguoya et al., 2017*). We report here that variants of HIV-1C that acquire novel late domain II-like, tetrapeptide insertions display ALIX binding to Gag-p6, enhanced virion release, CCL2-responsiveness and improved replication capacity of such viruses. Thus, we demonstrate how HIV-1 can modulate its own replication fitness via its genetic plasticity in maintaining, deleting or reacquiring the second late domain in Gag thus varying their ability to respond to CCL2. Our finding that CCL2 leads to redistribution of ALIX can have wide-ranging relevance to many other viruses and potentially in MVB and exosome biogenesis.

## Results

### CCL2 enhances clade B, but not clade C, HIV-1 virus particle release

It has been previously reported that CCL2 immuno-depletion inhibits HIV-1B virus production (*Fantuzzi et al., 2003*). To examine whether this phenomenon applies to HIV-1C (which does not induce CCL2 upon MDM infection), we tested the effect of CCL2 immuno-depletion on HIV-1C$_{IndieC1}$ replication and particle release in monocyte-derived macrophages (MDMs) in vitro and compared it to the effect on HIV-1B$_{ADA}$ replication. Immuno-depletion of CCL2 led to a significant reduction (~10 fold) in the replication of HIV-1B$_{ADA}$ (*Figure 1A*), accompanied by a corresponding increase in the intracellular accumulation of HIV-1 Gag precursor (pr55: *Figure 1C*). In contrast, immuno-depletion of CCL2 in HIV-1$_{IndieC1}$-infected MDMs neither inhibited virus replication nor increased intracellular Gag levels (*Figure 1B and C*). Comparison of relative amounts of p24 released into the culture medium as virus particles to intracellular levels of Gag suggested that virion release of HIV-1B$_{ADA}$, but not HIV-1C$_{IndieC1}$, is severely reduced upon CCL2 immuno-depletion (*Figure 1C*). Furthermore, addition of CCL2 to the HIV-1B$_{ADA}$-infected macrophage cultures enhanced virus replication, while HIV-1$_{IndieC}$–infected cells were unaffected (*Figure 1A and B*). Depletion or elevation of CCL2 in spent media was verified by measuring CCL2 levels in both macrophage and CEM-CCR5 culture media (*Figure 1—figure supplement 1A, B and C*). The CCL2-mediated stimulation of HIV-1 replication (~50% increase) was also observed in CEM-CCR5, a CD4$^+$ T-cell-derived cell line (*Figure 2*) but the α-CCL2 suppression of virus production was minimal (~25%). Together, our results indicated that particle release of HIV-1B, but not HIV-1C, was affected by modulating CCL2 levels in the medium. However, it appeared that the inhibition of virus replication by immune-depletion of CCL2 in CEM-CCR5 cells was much milder than in macrophages. To understand the basis for the

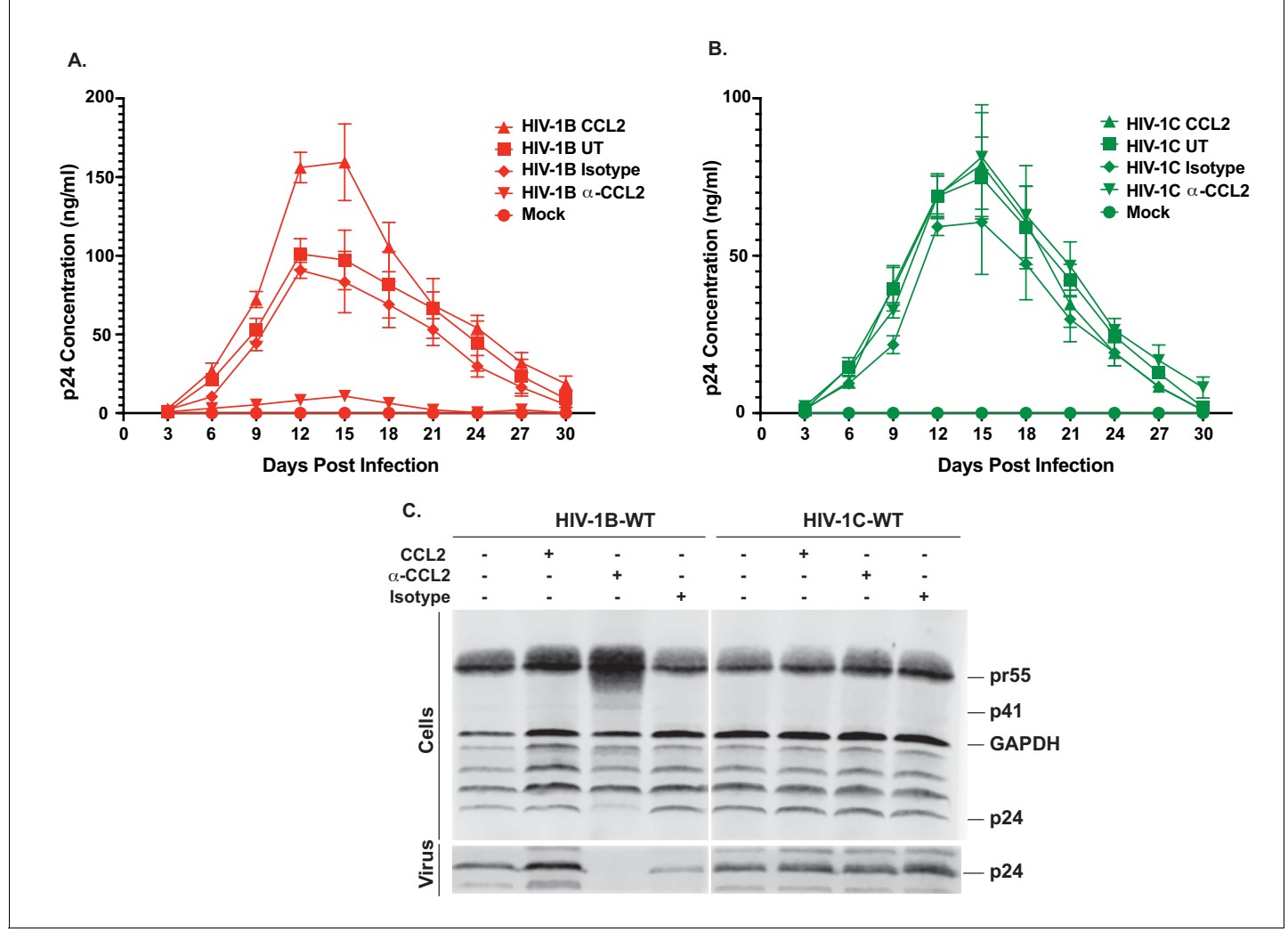

**Figure 1.** Differential effects of CCL2 levels in the medium on HIV-1B and HIV-1C replication in MDMs. (**A, B**) MDMs were infected with 10ng/1×10⁶ cells each of either HIV-1B$_{ADA}$ (panel A) or HIV-1C$_{IndieC1}$ (panel B) and the cultures propagated in the presence of CCL2 (0.25 µg/ml), α-CCL2 (2.5 µg/ml) or isotype antibody (2.5 µg/ml). Mock indicates no virus added and UT refers to no CCL2 or antibodies. Data are represented as mean ± SEM (n = 3). The experiment was performed three times with MDMs from three separate donors with two replicates per experiment. Means ± SEM are plotted. (**C**) Media and cells were collected at day 15, virus in the supernatant was pelleted *via* 20% sucrose cushion. Both the cell lysate and purified virus were analyzed in a Gag-p24 Western immunoblot.

DOI: https://doi.org/10.7554/eLife.35546.002

The following source data and figure supplements are available for figure 1:

**Source data 1.** All p24 values plotted in *Figure 1A* and *Figure 1B*.
DOI: https://doi.org/10.7554/eLife.35546.006

**Figure supplement 1.** CCL2 measurement in HIV-1 infected MDM, CEM-CCR5 culture media and HeLa cells.
DOI: https://doi.org/10.7554/eLife.35546.003

**Figure supplement 1—source data 1.** CCL2 values plotted in *Figure 1—figure supplement 1A and B*.
DOI: https://doi.org/10.7554/eLife.35546.007

**Figure supplement 1—source data 2.** CCL2 values plotted in *Figure 1—figure supplement 1C*.
DOI: https://doi.org/10.7554/eLife.35546.008

**Figure supplement 1—source data 3.** CCL2 values plotted in *Figure 1—figure supplement 1D*.
DOI: https://doi.org/10.7554/eLife.35546.009

**Figure supplement 2.** Assessing expression levels of CCR2b mRNA and protein in cells employed in this study.
DOI: https://doi.org/10.7554/eLife.35546.004

**Figure supplement 2—source data 4.** Fold change in CCR2b mRNA plotted in *Figure 1—figure supplement 2A*.
DOI: https://doi.org/10.7554/eLife.35546.010

*Figure 1 continued on next page*

*Figure 1 continued*

**Figure supplement 3.** Inhibition of virus replication by α-CCL2 can be reversed by exogenous CCL2.

DOI: https://doi.org/10.7554/eLife.35546.005

**Figure supplement 3—source data 5.** All the p24 values plotted in *Figure 1—figure supplement 3*.

DOI: https://doi.org/10.7554/eLife.35546.011

differential suppression by α-CCL2 between macrophages and CD4 T cells, we compared CCL2 protein levels in the spent media of macrophages, CEM-CCR5 cells and HeLa cells, a cell line that is frequently used in studies of HIV-1 late events (*Figure 1—figure supplement 1*). In the same three cell types, we also measured CCR2 mRNA and performed western blots to detect CCR2 protein (*Figure 1—figure supplement 2*). Our results indicated that robust levels of CCR2 mRNA and protein were detected in all three cell types. CCL2 production in both MDMs and HeLa cells was significant (*Figure 1—figure supplement 1A, B and D*), but the levels of CCL2 produced in CEM-CCR5 cells was below the level of detection – and we only detected miniscule amounts upon HIV infection or robustly only in samples where CCL2 was extraneously added (*Figure 1—figure supplement 1C*). These data suggest that the limiting amounts of CCL2 is primarily responsible for the minimal inhibition in virus replication observed upon the addition of α-CCL2 in this CD4 T cell line. Nevertheless, CCL2-mediated enhancement of virus production (conversely, inhibition of virus production by CCL2 depletion) was observed in both macrophages and T cells, indicating that this effect is not limited to macrophages. Importantly, CCR2 is present on both CD4 T cells and macrophages, suggesting that both can respond to the presence of CCL2 by increasing virus replication.

To verify that the HIV-1B virus inhibition observed in the presence of α-CCL2 was due to the neutralization of CCL2, and not due to an off-target effect, we performed a rescue experiment by adding back CCL2 after CCL2 immuno-depletion. A multi-day macrophage HIV-1B$_{ADA}$ infection experiment was performed as in *Figure 1*, where, on the 9th day, α-CCL2 antibodies in the medium were replaced with CCL2 and virus replication was monitored up to 24 days post-infection. Our

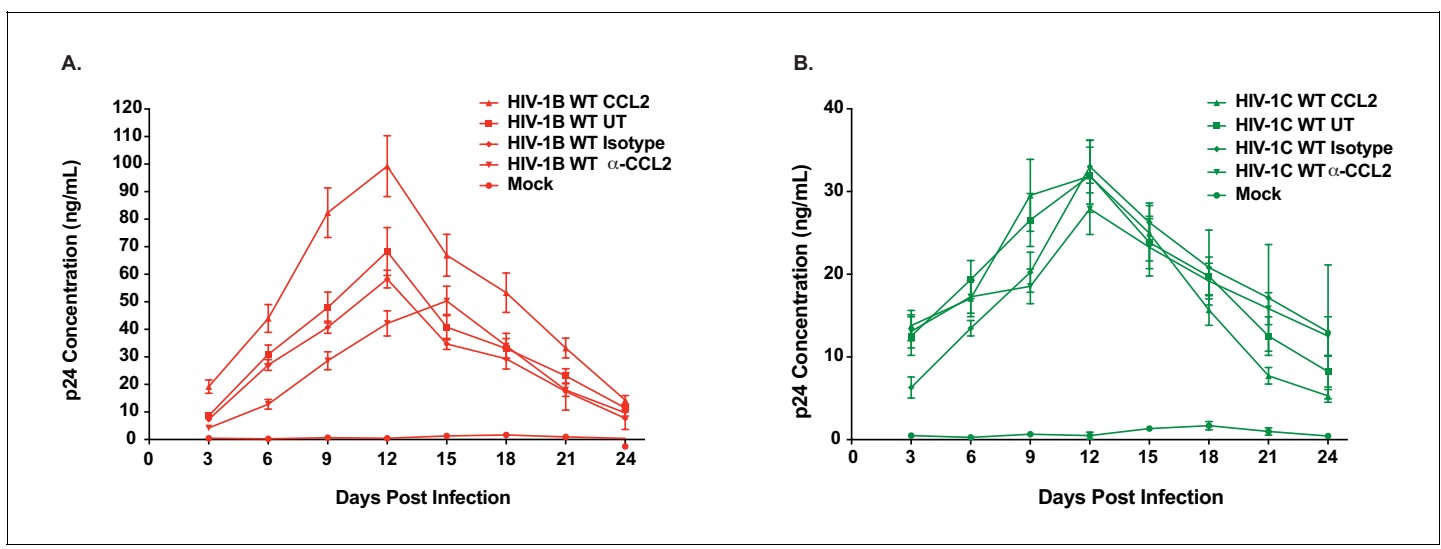

**Figure 2.** Differential CCL2-responsiveness of HIV-1B and HIV-1C is also observed in a T cell line albeit to a lower degree. (A, B) CEM-CCR5 cells were infected with 10ng p24/ 1×10$^6$ cells input of either HIV-1B$_{ADA}$ (panel A) or HIV-1C$_{Indiec1}$ (panel B) and the culture was propagated in the presence of no additional treatment (UT) , CCL2 (0.25 µg/ml), α-CCL2 (2.5 µg/ml) or isotype antibody (2.5 µg/ml). Mock indicates no virus added and UT refers to no CCL2 or antibodies. Data are represented as mean ± SEM (n = 3). The experiment was performed three times.

DOI: https://doi.org/10.7554/eLife.35546.012

The following source data is available for figure 2:

**Source data 1.** All p24 values plotted in *Figure 2A* and *Figure 2B*.

DOI: https://doi.org/10.7554/eLife.35546.013

results showed that while the virus replication was inhibited until day 9, it resumed at day 9 and reached peak virion production at day 18 (*Figure 1—figure supplement 3*).

## Lack of a LYPX motif in HIV-1 C disrupts ALIX binding to Gag-p6

It has been reported that HIV-1C isolates contain a LY dipeptide deletion in the Gag-p6 LYPX motif, whereas all non-HIV-1C isolates examined retain an intact LYPX motif (*Patil and Bhattacharya, 2012*). To determine the extent of the absence of the LYPX motif in HIV-1 clade C isolates world-wide, we performed Clustal-X analysis of Gag p6 sequences of HIV-1B and HIV-1C prevalent in different geographic regions from the Los Alamos database and examined the conservation of both PTAP and LYPX motifs. Our HIV-1B vs. HIV-1C alignment shows that while LYPX motif was conserved in clade B sequences, it was not conserved in clade C sequences, mainly due to a LY dipeptide deletion (*Figure 3A*). Previous studies have revealed that the tyrosine 36 (Y36) residue in the late motif II (LYPX) of Gag is critical for binding to the conserved F676 in the flexible 'V' domain of ALIX (*Fisher et al., 2007*; *Lee et al., 2007*; *Fujii et al., 2009*). Our analysis of 695 clade B and 475 clade C isolates showed that while the Y36 residue is conserved in about 96% of clade B isolates examined, it was absent in >99% of clade C isolates examined (*Figure 3B*). On the other hand, the PTAP motif was fully conserved in both B and C isolates. To confirm that the absence of LYPX motif in HIV-1 C Gag-p6 disrupts ALIX-binding, we examined the ability of Gag-p6 from HIV-1B$_{ADA}$ and HIV-1C$_{IndieC1}$ viruses to bind ALIX using in vitro GST pull-down assay. Our results indicated that HIV-1B, but not HIV-1C Gag-p6 bound to ALIX (*Figure 3C*).

## CCL2-responsiveness is dependent on the presence of LYPX motif in HIV Gag-p6 and ALIX in the cell

Our data above demonstrated that the absence of CCL2 leads to intracellular accumulation of Gag only in HIV-1B infection. Since the two clades differ in the presence or absence of LYPX motif in Gag-p6 (*Figure 3*, *Figure 4A*), and since our results demonstrated that CCL2 differentially influences virus replication of the two clades (*Figure 1*, *Figure 2*), we surmised that CCL2-mediated virus release was dependent on the LYPX, and not on the PTAP motif. To test this hypothesis, we created a panel of HIV-1B$_{ADA}$ molecular clones bearing mutations either in LYPX (by deleting the LY dipeptide) or in PTAP (by substituting with LIRL) (*Huang et al., 1995*) or both. The mutant viruses were tested for their ability to respond to CCL2 addition or depletion in HeLa cells that were transfected with molecular clones of each of the mutants. As indicated before (*Figure 1*), we found that CCL2 addition increased, and CCL2 depletion inhibited the virion particle production of HIV-1B wild type (*Figure 4B*). We found that deletion of the PTAP motif drastically reduced (by ~75%) virus production (reflecting the well-known major role of the PTAP motif in virus budding). Interestingly, even though the virus production was significantly reduced due to the absence of PTAP, HIV-1B$_{ADA}$$^{-PTAP (+LIRL)}$ showed CCL2 responsiveness (*Figure 4B*, -PTAP). Deletion of the LY dipeptide (ΔLY mutation) in HIV-1B showed a less severe reduction in virus production, reflecting the auxiliary role of the LYPX motif (*Figure 4B*). Importantly, LY deletion in HIV-1B completely eliminated the CCL2-responsiveness as neither the addition nor depletion of CCL2 had any effect on virus production of HIV-1B$_{ADA}$$^{ΔLY}$ (*Figure 4B*). In fact, the pattern of virus production of HIV-1B$_{ADA}$$^{ΔLY}$ was identical to that of wild-type HIV-1C$_{IndieC1}$ in all three conditions (*Figure 4B and C*). When both motifs were absent, virus production was completely abrogated (*Figure 4B*). These results indicated that CCL2-responsiveness was dependent on the presence of LY motif in HIV-1.

We next generated a corresponding panel of mutants in HIV-1C$_{IndieC1}$ by deleting PTAP (HIV-1C$_{IndieC1}$$^{-PTAP (+LIRL)}$) or creating an LYPX motif by inserting an LY dipeptide (HIV-1C$_{IndieC1}$$^{+LY}$) or creating both mutations together in one clone (HIV-1C$_{IndieC1}$$^{-PTAP (+LIRL), +LY}$) and tested their responsiveness to depletion or addition of CCL2 (*Figure 4C*) upon transfection to HeLa cells. We found that the insertion of the LY dipeptide (HIV-1C$_{IndieC1}$$^{+LY}$) rendered HIV-1C$_{IndieC1}$ responsive to the presence or absence of CCL2 and yielded a pattern of viral production that recapitulated the wild-type HIV-1B$_{ADA}$ (*Figure 4C*) irrespective of whether or not there was an intact PTAP motif (HIV-1C$_{IndieC1}$$^{+LY}$ and HIV-1C$_{IndieC1}$$^{-PTAP (+LIRL), +LY}$). Furthermore, wild-type virus, which lacks a LYPX motif (HIV-1$_{IndieC1}$) was insensitive to changes in CCL2 levels in the medium. The virus lacking both late motifs (HIV-1$_{IndieC1}$$^{ΔPTAP (LIRL)}$) produced little virus. These results together, for the first time,

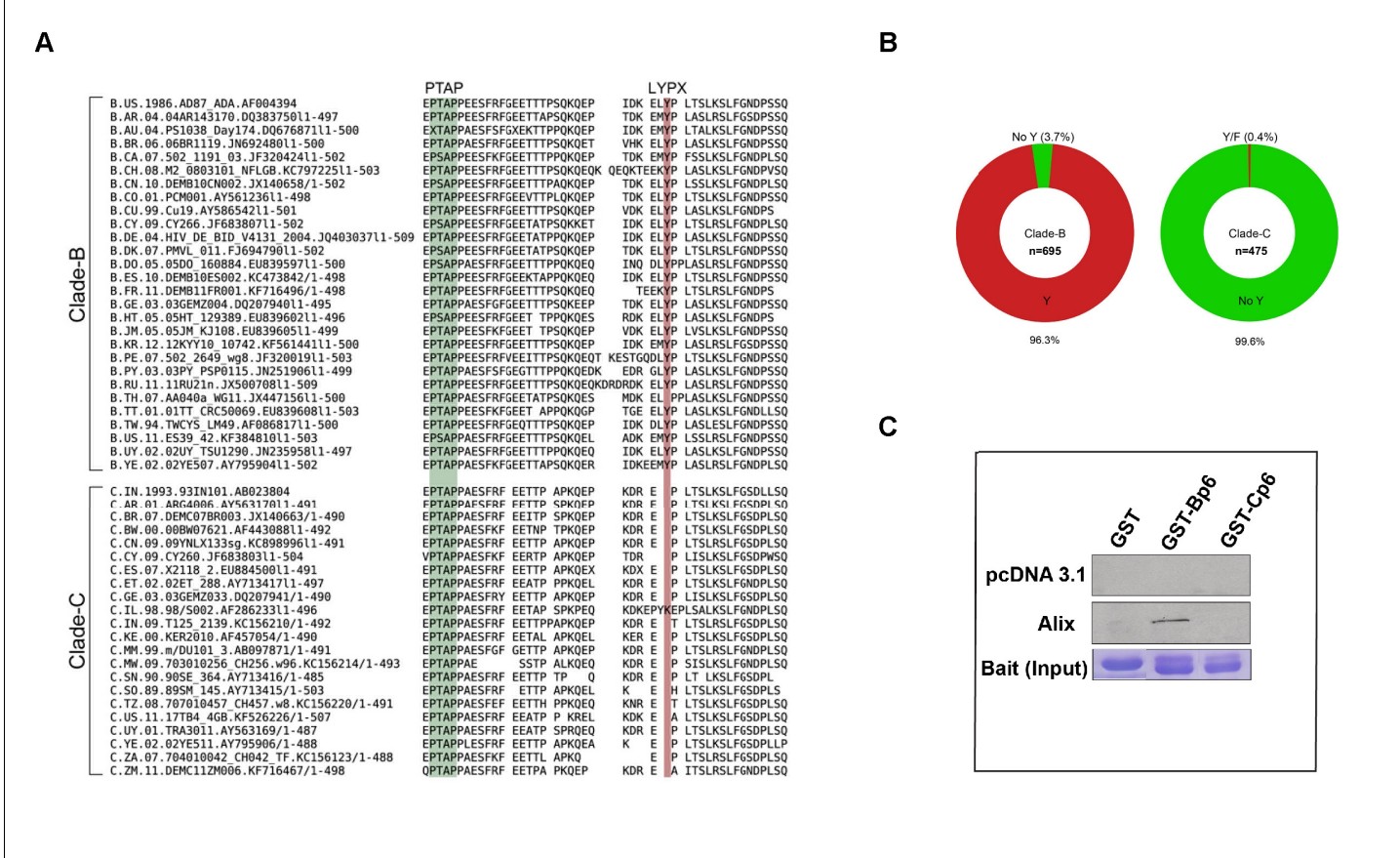

**Figure 3.** Sequence alignment of HIV-1B and HIV-1C Gag-p6 proteins, and Gag p6-ALIX interaction. (**A**) Representative HIV-1 Gag p6 sequences from either HIV-1B or HIV-1C isolates that are prevalent in different geographic regions were selected from the Los Alamos database and aligned using Clustal-X. As shown, HIV-1B Gag-p6 retains both the late domains (PTAP and LYPX$_n$L), whereas HIV-1C Gag p6 sequences have a LY deletion. (**B**) The pie charts represent the frequencies of clade B (left) or clade C (right) sequences with or without the residue Y (of the LYPX motif) in Gag-p6. Clade-B sequences containing the Y residue are shown in red, the remaining sequences have been combined and shown in green. The frequencies of residues replacing Y are 1.6% (**H**), 0.9% (none), 0.4% (**C**), 0.3% (**F**), 0.3% (**S**), 0.1% (**N**) and 0.1% (**P**) respectively. Clade-C sequences without a Y (LY deletion) are shown in green and the 0.2% of the sequences that retain the Y residue or the 0.2% sequences that contain an F residue are combined and shown in red. The total number of sequences examined in Clade-B or Clade-C are 695 and 475, respectively, after removing redundancy at 95%. (**C**) Employing lysates of Hela cells transfected with either empty vector (pcDNA 3.1) or ALIX-expressing pcDNA construct, pull-downs were performed using GST by itself or GST fused to either HIV-1B-p6 or HIV-1C-p6. GST-B-p6 binds to ALIX, whereas GST-C-p6 does not, due to absence of LY in the HIV-1C Gag p6.

DOI: https://doi.org/10.7554/eLife.35546.014

The following source data is available for figure 3:

**Source data 1.** HIV-1 clade C sequences used in the analysis in *Figure 3B*.
DOI: https://doi.org/10.7554/eLife.35546.015
**Source data 2.** HIV-1 clade B sequences used in the analysis in *Figure 3B*.
DOI: https://doi.org/10.7554/eLife.35546.016
**Source data 3.** Sequences that were aligned in *Figure 3A*.
DOI: https://doi.org/10.7554/eLife.35546.017

demonstrated that in both HIV-1 clades B and C, CCL2 effects are mediated by the LYPX, but not by the PTAP motif.

To determine the equivalence of intracellular Gag in the various treatments for all the mutant viruses and if the decrease in particle production in various conditions in the above experiments was not due to a reduction in the intracellular Gag levels, we carried out an immunoblot analysis of the cell lysates using α-p24 antibodies. As a loading control, we used α-GAPDH antibodies (*Figure 4— figure supplement 1*). Our results indicated that a decrease in extracellular p24 was associated with

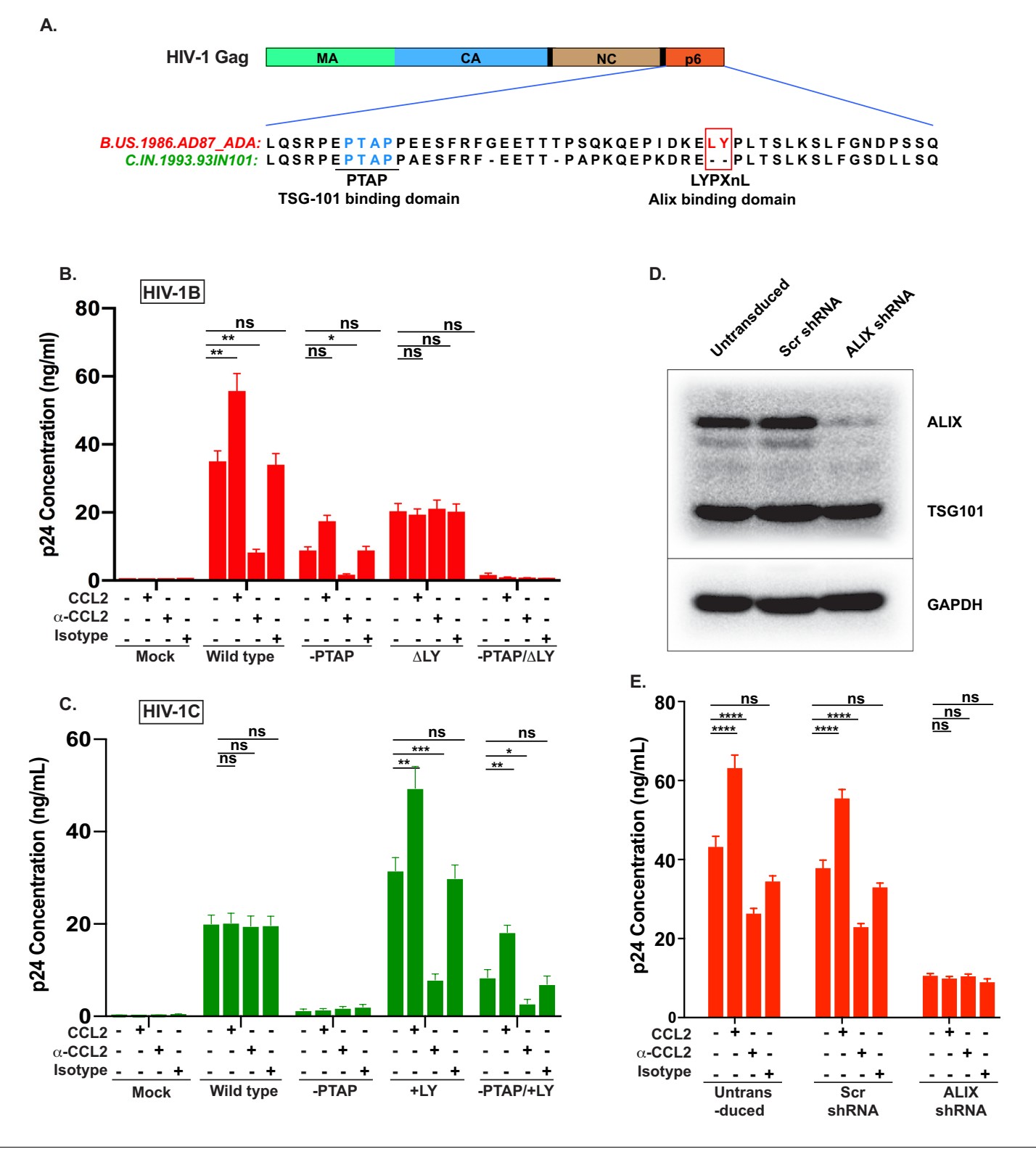

**Figure 4.** CCL2-mediated changes in virus production requires the presence of Gag-p6 LYPX motif in the virus and ALIX in target cells. (**A**) Alignment of Gag-p6 amino acid sequences from the two viruses employed in this study - HIV-1B_ADA and HIV-1C_IndieC1. HIV-1B_ADA has both late motifs (PTAP and LYPX) while HIV-1C_IndieC1 lacks the LY dipeptide in LYPX motif. (**B, C**) Modulation of virus production by CCL2 levels in the media of HeLa cells transfected with HIV-1B_ADA (panel B) or HIV-1C_IndieC1 (panel C) molecular clones or their late domain mutant versions. (**B**) Shows results of transfecting

*Figure 4 continued on next page*

*Figure 4 continued*

HIV-1B$_{ADA}$ or its -PTAP, ΔLY or –PTAP/ΔLY variants and (**C**) Shows HIV-1C$_{IndieC1}$ or its –PTAP, +LY or –PTAP/+LY variants. For each virus, p24 production 24 hr post transfection is plotted including with no treatment, in the presence of CCL2 (0.25 μg/ml), α-CCL2 antibody (2.5 μg/ml) or Isotype antibody (2.5 μg/mL). Data are represented as mean ± SEM (n = 4). The experiment was performed four times with two replicates in each experiment. (**D**) CEM-CCR5 cells were transduced with lentiviral vectors carrying either a control scrambled shRNA or an *ALIX* shRNA 29394 expression cassette. Western blot analysis of lysates of untransduced CEM-CCR5 cells or transduced with either the scrambled shRNA or the *ALIX* shRNA is shown. The bottom panel shows GAPDH protein as a loading control. (**E**) CEM-CCR5 cells with *ALIX* knocked-down were infected with HIV-1B$_{ADA,}$ either in the presence of CCL2, α-CCL2 or Isotype antibody followed by measurement of virus replication at 12 days post-infection. The results show that CCL2-responsiveness of HIV-1B$_{ADA}$ is dependent on *ALIX* expression. The experiment was performed three times and the mean ± SEM are plotted.

DOI: https://doi.org/10.7554/eLife.35546.018

The following source data and figure supplements are available for figure 4:

**Source data 1.** All p24 values plotted in *Figure 4B* and 4C.
DOI: https://doi.org/10.7554/eLife.35546.021
**Source data 2.** All p24 values plotted in *Figure 4E*.
DOI: https://doi.org/10.7554/eLife.35546.022
**Figure supplement 1.** Western blot to measure intracellular capsid protein levels.
DOI: https://doi.org/10.7554/eLife.35546.019
**Figure supplement 2.** Effect of CCL2 levels on the infectivity of HIV-1B, HIV-1C and L-domain mutant viruses.
DOI: https://doi.org/10.7554/eLife.35546.020
**Figure supplement 2—source data 1.** Relative ight units (from Luciferase assays) plotted in *Figure 4—figure supplement 2*.
DOI:

a corresponding intracellular increase in Pr55 Gag levels, specifically in a manner dependent on the presence of LYPX motif (see α-CCL2 conditions for HIV-1B WT and HIV-1C$^{+LY}$, HIV-1C$^{-PTAP, +LY}$), establishing that the inhibition of particle production was not due to an inherent decrease in intracellular Gag, but rather due to an assembly and/or particle release step defect (*Figure 4—figure supplement 1*).

To determine if the effects of CCL2 were mediated by modulating early events, all mutants in both HIV-1 clades were tested for infectivity using equivalent input (10ng p24 each) of viruses in parallel infections of the reporter cell line TZMbl and measuring infectivity by Luciferase activity (*Figure 4—figure supplement 2*). While the PTAP mutation reduced infectivity significantly in both HIV-1$_{ADA}$ and HIV-1$_{IndieC1}$, the LY deletion in HIV-1$_{ADA}$ had a 30% infectivity defect and the LY insertion in HIV-1$_{IndieC1}$ had no effect. Double mutations in both cases had a severe loss of infectivity. However, we did not see any effect of CCL2 addition or depletion on the infectivity of the viruses, suggesting that the CCL2-dependency manifests at the late events but not at the early events (*Figure 4—figure supplement 2*).

In summary, measurement of virus production under various treatments indicated that viruses that harbored an LYPX motif (HIV-1$_{ADA}$ WT, HIV-1$_{ADA}$$^{-PTAP (+LIRL)}$, HIV-1C$_{IndieC1}$$^{+LY}$ and HIV-1C$_{IndieC1}$$^{-PTAP +LY}$) responded to CCL2 modulation and viruses that lacked the LYPX motif (HIV-1$_{ADA}$$^{PTAP, ΔLY}$ and HIV-1C$_{IndieC1}$ WT), irrespective of the presence or absence of PTAP, were non-responsive to CCL2 (*Figure 4B and C*).

Since the main function of the LYPX motif is to recruit ALIX, we surmised that the effects of CCL2 are mediated by the ability of Gag-p6 to bind ALIX. To test this hypothesis, we examined the role of ALIX in CCL2-responsiveness by silencing the *ALIX* gene (*PDCD6IP)* in HIV producer cells and determined the effect on CCL2-responsiveness of viruses harboring the LYPX motif. *ALIX* was knocked down in CEM-CCR5 cells via shRNA. Immunoblot analysis indicated that the knock-down resulted in a > 95% decrease in cellular ALIX protein (*Figure 4D*). We infected these cells with wild type HIV-1B$_{ADA}$ virus, in the presence of CCL2, α-CCL2 or an isotype antibody and measured virus production at day 15 post-infection. In the α-CCL2 condition, there was a reduction in the level of virus production, although not as much as seen in macrophages (compare *Figure 4E* to *Figure 1*) or HeLa cells (compare *Figure 4E* to *Figure 4B*). Virus production was reduced by about threefold in *ALIX* knock-down cells and was not further influenced by the presence or absence of CCL2 (*Figure 4E*). These results collectively established that the responsiveness to CCL2 requires LYPX motif in the virus and the presence of ALIX in the cell.

## CCL2 increases soluble ALIX and facilitates Gag-p6 binding

In view of the contribution of Gag-p6 binding to ALIX toward virion budding and release, and based on the fact that CCL2 responsiveness is dependent on the presence of ALIX (*Figure 4E*), we hypothesized that CCL2 may mediate its effect by inducing ALIX expression. We performed a quantitative reverse transcriptase-polymerase chain reaction (qRT-PCR) to determine the *ALIX* mRNA levels in HeLa cells treated with CCL2 (0.25 μg/ml) or α-CCL2 antibodies (2.5 μg/ml). RT-qPCR analysis of mRNA in HeLa cells over multiple time points indicated that *ALIX* mRNA levels remained unaltered in the presence or absence of CCL2 or α-CCL2 (*Figure 5A*).

We next examined ALIX protein levels. We considered that while total ALIX levels may remain the same, the accessible protein levels (soluble ALIX) may be affected by CCL2 levels. Therefore, we examined the total, soluble and insoluble fractions of ALIX in untreated, CCL2-treated and α-CCL2-treated HeLa cells. As shown in *Figure 5B*, total ALIX levels remained the same with all the treatments, consistent with the qRT-PCR results (*Figure 5B* top panel of immunoblots and the corresponding graph – blue bars - to the right). However, the level of soluble ALIX in untreated or CCL2-treated cells was higher than in α-CCL2-treated cells (*Figure 5B* – middle panel: 4 hr treatment; compare lanes 1 and 2 to 3, UT and CCL2 to α-CCL2, and the corresponding bar graphs, middle panel). There was very little insoluble ALIX (Pellet 2) with CCL2-treatment (*Figure 5B*, bottom panel, lane 2), whereas increased ALIX was detected in the insoluble fraction upon α-CCL2 treatment (*Figure 5B*, bottom panel, lane 3 – see corresponding bar graph). These results indicated that while total cellular ALIX remained the same under all three conditions (top row, lanes 1 to 4), soluble ALIX was increased by CCL2 (middle row, lane 2).

We next examined the formation of Gag-p6-ALIX complexes in the presence and absence of CCL2, using recombinant glutathione S- transferase (GST) fusions of Gag-p6 proteins of HIV-1B$_{ADA}$ and HIV-1C$_{indiec1}$ viruses. Lysates from untreated, CCL2- or α-CCL2-treated HeLa cells were incubated with GST, GST-Bp6 or GST-Cp6 proteins respectively, immobilized on glutathione beads and the bound complexes were subjected to immunoblot analysis. There was binding of GST-Bp6 to ALIX under untreated condition (*Figure 5C*, lane 4). GST-Bp6 bound to more ALIX when CCL2 was added to the medium, which was consistent with an increase in soluble ALIX in the cells (*Figure 5C*, compare lanes 4 and 5). However, under conditions where CCL2 was depleted (α-CCL2), no association was detected (*Figure 5C*, compare lanes 4 and 6). Soluble ALIX was detected in both untreated (lane 4) and CCL2-treated (Lane 5) cell lysates (inputs) but was undetectable in the α-CCL2-treated cell lysates (Lane 6). As expected, ALIX did not bind GST-Cp6 under any condition (*Figure 5C*, lanes 7–9) despite the presence of robust levels of ALIX in the soluble fraction (*Figure 5C*, input). As a control, we also tested the association of TSG101 with GagB-p6 and GagC-p6. We found that unlike ALIX, p6-binding to TSG101 was not affected by changing CCL2 levels (*Figure 5*). Both GagB-p6 and GagC-p6 proteins bound to TSG101 in all cases, suggesting that responsiveness to CCL2 is specifically correlated to ALIX-binding but not TSG101-binding (*Figure 5C*). These results indicate that: (i) intracellular Gag-p6-ALIX binding is dependent on the presence of the LYPX motif in Gag-p6 and that this interaction can be modulated by the presence of CCL2 in the medium of HIV-infected cells; and (ii) HIV-1C Gag-p6 is unable to bind to ALIX irrespective of CCL2 addition or depletion.

## CCL2 mobilizes ALIX from actin cytoskeletal structures

It is known that CCL2 significantly reorganizes the actin cytoskeleton in microglia (*Cross and Woodroofe, 1999*). ALIX is also intimately involved in actin rearrangement and polymerization in fibroblasts by directly interacting with F-actin through both the Bro1 and the proline-rich domains (*Pan et al., 2006*). Furthermore, ALIX knockdown affects both the cellular levels and distribution of F-actin (*Bongiovanni et al., 2012*). We surmised that the CCL2-dependent increase in soluble ALIX may involve mobilization of ALIX from the F-actin cytoskeleton to the cytosol, rendering it accessible to Gag-p6. To test this hypothesis, we used confocal microscopy to examine ALIX distribution in HeLa cells and macrophages under conditions that modulate CCL2 levels. In HeLa cells, our results indicated that when CCL2 was immunodepleted (α-CCL2 treatment), ALIX was strictly associated with F-actin stress fibers (*Figure 6A*, 'α-CCL2' row - compare ALIX, Actin and Merged panels). In contrast, in CCL2-treated samples (including untreated cells, which contain endogenous CCL2, or cells treated with exogenous CCL2), ALIX was not associated with F-actin structures and was mostly distributed in the cytoplasm (*Figure 6A*, 'CCL2' row - compare ALIX, Actin and Merged panels).

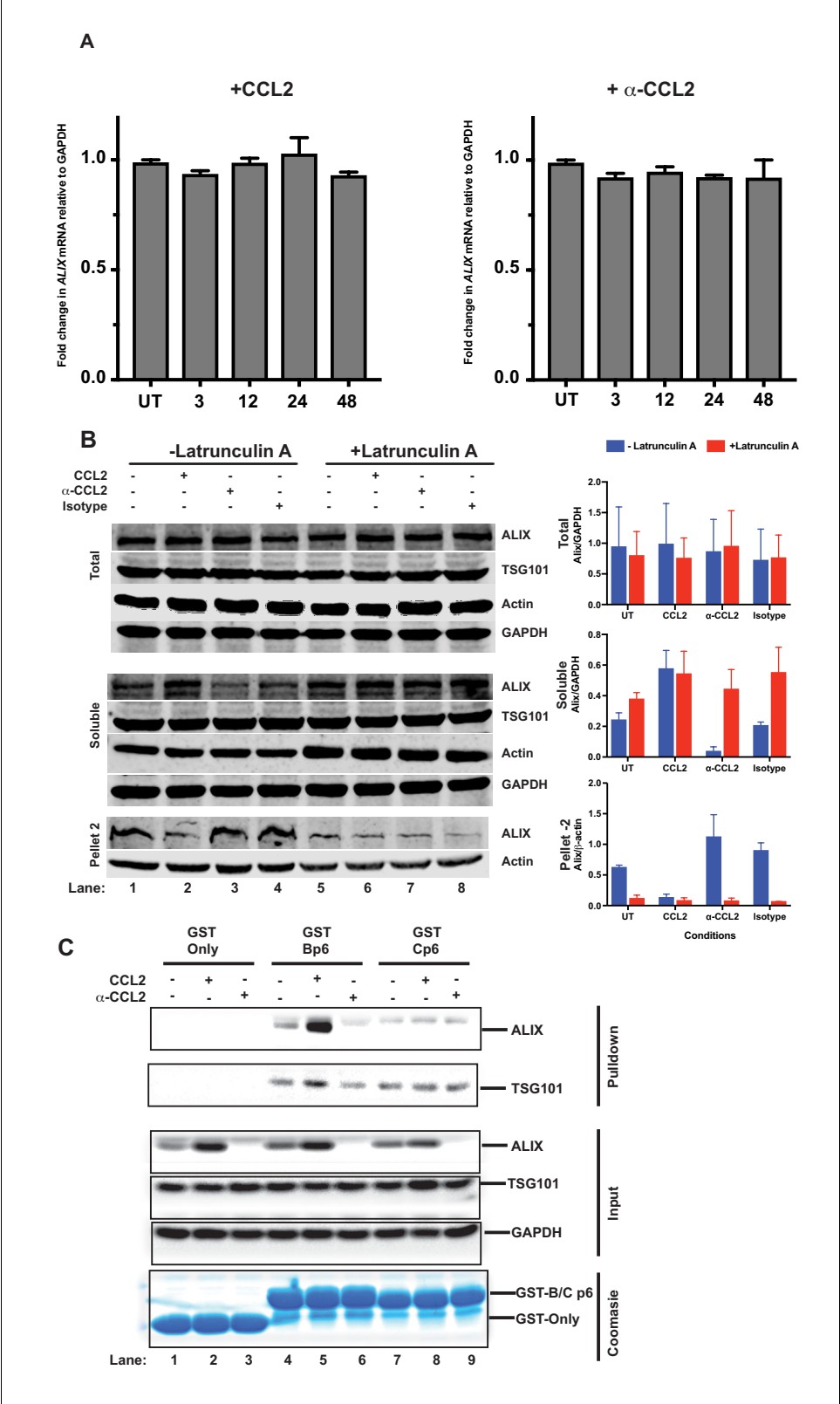

**Figure 5.** CCL2 does not impact *ALIX* mRNA levels but increases soluble ALIX and facilitates Gag-p6 binding. (**A**) Effect of CCL2 levels on *ALIX* mRNA levels. Total RNA from HeLa cells treated with CCL2 (0.25 µg/ml) or α-CCL2 (2.5 µg/ml) was extracted, reverse transcribed and the cDNAs employed in real-time polymerase chain reactions. Data are represented as mean ± SEM (n = 3). The experiment was performed three times. (**B**) Immunoblots showing ALIX levels in the total, soluble and insoluble fractions of HeLa cells. Total ALIX protein levels in untreated (UT), CCL2 treated or α-CCL2-

*Figure 5 continued on next page*

*Figure 5 continued*

treated HeLa cells are similar (*Top panel*). However, upon CCL2 treatment, most of the ALIX is in the soluble phase while α-CCL2 treatment shows the reverse (*middle panel*) ALIX levels in pellet II (which constitutes uncross-linked actin - see Materials and methods) represent the converse of what is seen in the soluble fraction (*bottom Panels*). Two identically treated batches of cells were used, but with or without Latrunculin treatment. ALIX band intensities are quantified, normalized for GAPDH (Total and Soluble) or Actin (Pellet 2) and plotted to the right of the immunoblot. The experiment has been performed three times. (**C**) CCL2 treatment facilitates Gag-p6-ALIX interaction. GST protein, either by itself or fused to HIV-1B-p6 or HIV-1C-p6 proteins, were expressed in *E. coli* and purified using Glutathione agarose beads, and incubated with HeLa cell lysates that were either untreated, treated with CCL2 or with α-CCL2 for 24 hr followed by pull-down experiments (two *top panels*). The input control panels show that the quantities of all proteins (GST and its p6 fusions, ALIX, TSG101 and GAPDH) in the cell lysates from all treatments were comparable within each set – except ALIX, which appears to be absent upon α-CCL2 treatment (three *middle panels*). The *bottom panel* shows that the levels of GST bait proteins used were also at equivalent levels. The experiment was performed three times.

DOI: https://doi.org/10.7554/eLife.35546.024

The following source data is available for figure 5:

**Source data 1.** The mRNA fold change values for *Figure 5A*.
DOI: https://doi.org/10.7554/eLife.35546.025
**Source data 2.** Densitometric intensity values for ALIX bands in *Figure 5B*.
DOI: https://doi.org/10.7554/eLife.35546.026

Further magnification of sections of merged panels clearly shows that the colocalization is only seen in α-CCL2-treatment (*Figure 6A*, 'Zoom' column). We also examined ALIX localization in macrophages. Macrophages do not have stress fibers. Instead, actin is organized into other structures. One of the cytoskeletal structures are called podosomes (*Evans and Matsudaira, 2006*) - actin-rich degradative structures found on the ventral surface of adherent cells, and also in surfaces that project into the extracellular matrix. Although the ALIX distribution in untreated and CCL2 treated macrophages appeared more punctate than in HeLa cells, it was diffusely distributed throughout the cell and was not associated with the large punctate podosomes containing F-actin (*Figure 7A*, 'Untreated' and 'CCL2' rows – compare ALIX, Actin and Merged panels). In contrast, upon CCL2 depletion, ALIX completely colocalized with F-actin in podosome structures (*Figure 7A*, 'α-CCL2' row - compare ALIX, Actin and Merged panels). To further establish the colocalization of F-actin and ALIX, we determined Pearson colocalization coefficients for both HeLa cell and macrophage confocal images, which provides a quantitative measure of the degree of colocalization between two fluorophores (*Figure 6B* and *Figure 7B*). We found that the Pearson coefficient is highest when the cells were treated with α-CCL2 antibodies, indicating that in the absence of CCL2, ALIX is associated with actin cytoskeletal structures in the cytoplasmic compartment, while at basal level and increased CCL2 levels, it is maintained in the cytoplasm away from actin cytoskeleton.

Next, we investigated whether the association of ALIX with the actin cytoskeletal compartment diminishes upon F-actin depolymerization in HeLa cells using Latrunculin A, which binds monomeric G-actin molecules preventing their polymerization, with the net effect being F-actin depolymerization. HeLa cells were treated as before (untreated controls, CCL2-, α-CCL2-, and isotype antibody-treated cells). Following Latrunculin A treatment, we prepared detergent insoluble cytoskeletons, which are composed of crosslinked actin filaments, myosin-II and associated actin binding proteins (*Hartwig and Shevlin, 1986*; *Rosenberg et al., 1981*; *Tarone et al., 1984*). While detergent-resistant cytoskeletons contain crosslinked actin filaments, short actin filaments will remain in the supernatant. To evaluate the ALIX-actin interaction, we separated detergent-resistant F-actin cytoskeletons and soluble F-actin by low-speed centrifugation. The soluble fraction, which contains F-actin that is not crosslinked and actin monomers (G-actin), was centrifuged at high speed to pellet the soluble F-actin (labeled 'Pellet 2' in *Figure 5B*) and G-actin will remain in the supernatant. As expected, our results show that Pellet 2 contains F-actin, which is expected to be uncross-linked F-actin. In addition, we used Latrunculin A to reduce the amount of F-actin. Consistent with previous observations (*Peterson and Mitchison, 2002*), Latrunculin A treatment did not lead to a complete dissolution of all F-actin structures (*Figure 5B*, see actin bands in bottom panel – Pellet 2) although some reduction is observed. Interestingly, when the cells were treated with Lantrunculin A, the amount of ALIX found in the soluble and insoluble fractions with all the treatments remained same, consistent with the idea that disruption of actin cytoskeleton by Lantrunculin A allows ALIX to be

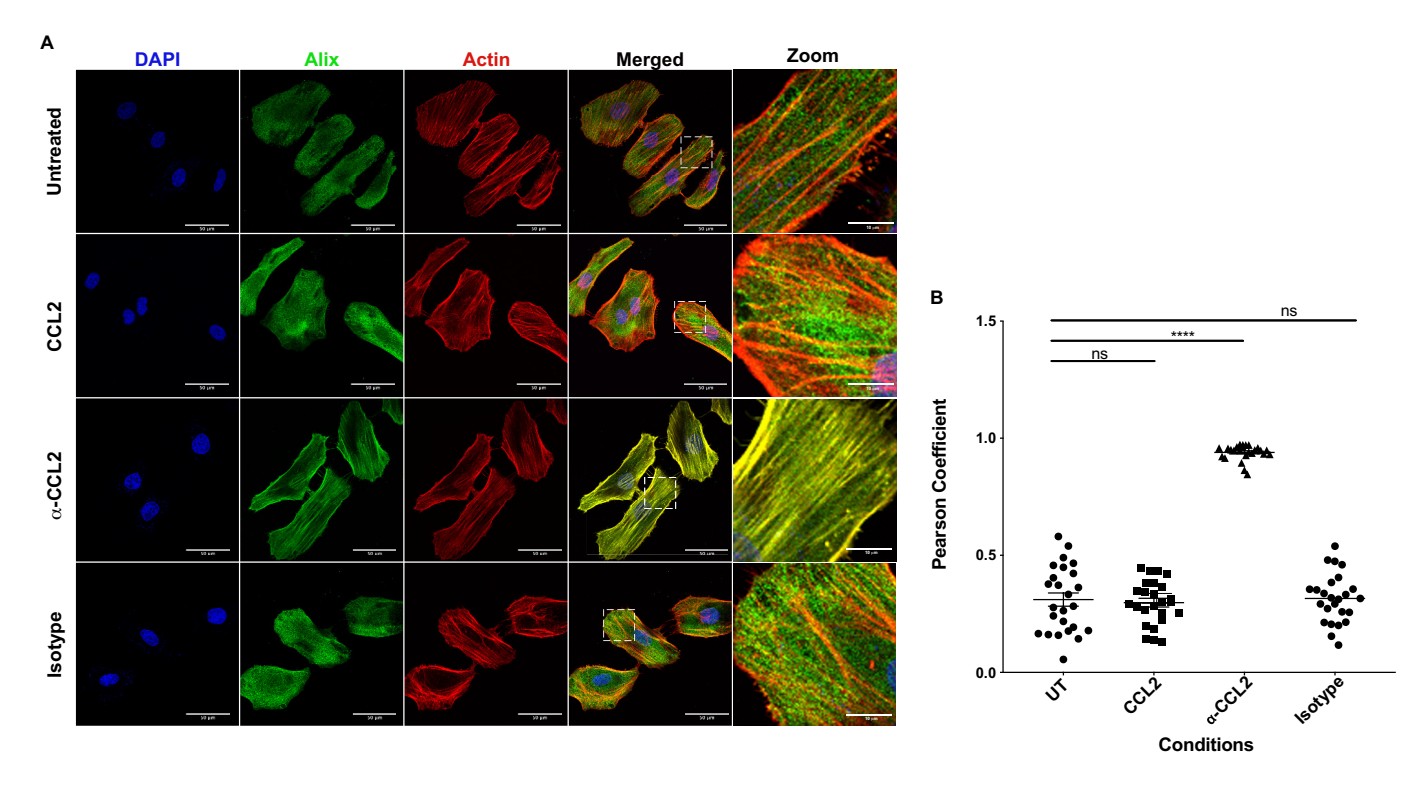

**Figure 6.** CCL2 triggers intracellular redistribution of ALIX in HeLa cells. (**A**) Confocal microscopy to visualize ALIX in HeLa cells. Cells treated with nothing (Untreated), CCL2, α-CCL2 or Isotype antibody for 4 hr were stained for ALIX, F-actin and DAPI to determine the cellular localization of ALIX upon modulating CCL2 levels. Fluorescence micrographs show that upon CCL2 depletion (α-CCL2 panels), ALIX is dramatically redistributed from the cytoplasm to F-actin stress fibers, while in the presence of CCL2 (both Untreated and CCL2-treated), ALIX is more diffuse throughout the cytoplasm and near the plasma membrane. A small portion of the merged panels was amplified 5.21X to show that the signals colocalize only in α-CCL2-treated cells (**B**) To quantify the degree of colocalization of ALIX and F-actin signals under conditions of CCL2 addition and depletion, we calculated the Pearsoncorrelation coefficients between the two fluorophores. For HeLa cells, the data plotted indicates the close proximity of F-actin and ALIX upon CCL2 depletion, while in the presence of CCL2 (untreated and CCL2) ALIX and F-actin signals are distinct from each other. Data are represented as mean ± SEM (n = 50 cells each for HeLa cells).

DOI: https://doi.org/10.7554/eLife.35546.027

The following source data is available for figure 6:

**Source data 1.** Pearson coefficient values plotted in *Figure 6B*.

DOI: https://doi.org/10.7554/eLife.35546.028

released (*Figure 5B*, middle and lower panel, lanes 5–8). These data provide strong support for CCL2-mediated regulation of ALIX's association with actin cytoskeleton.

Our results, for the first time, indicated that CCL2 regulates the redistribution of ALIX by releasing it from actin cytoskeleton, which in turn increases the levels of cytoplasmic ALIX accessible to Gag-p6 for binding. Our data also demonstrated a decreased availability of ALIX in the absence of CCL2 and suggested a mechanism for how CCL2 stimulates HIV-1B particle release.

## Effect of ALIX-actin colocalization on HIV-1 virus production

To understand the effect of ALIX sequestration to the actin cytoskeleton on HIV-1 release, we examined Gag distribution in HeLa cells transfected with the HIV-1B$_{ADA}$ molecular clone and that were treated with α-CCL2. Normally Gag traffics toward the plasma membrane in virus producer cells resulting in a Gag gradient with increased Gag near the plasma membrane as compared to the rest of the cytoplasm. Cano et al. have previously demonstrated that this gradient can be measured by quantifying the Gag pixel intensity in immunofluorescence images of HIV producer cells obtained by confocal microscopy following α-p24 antibody staining (*Cano and Kalpana, 2011*; *La Porte and*

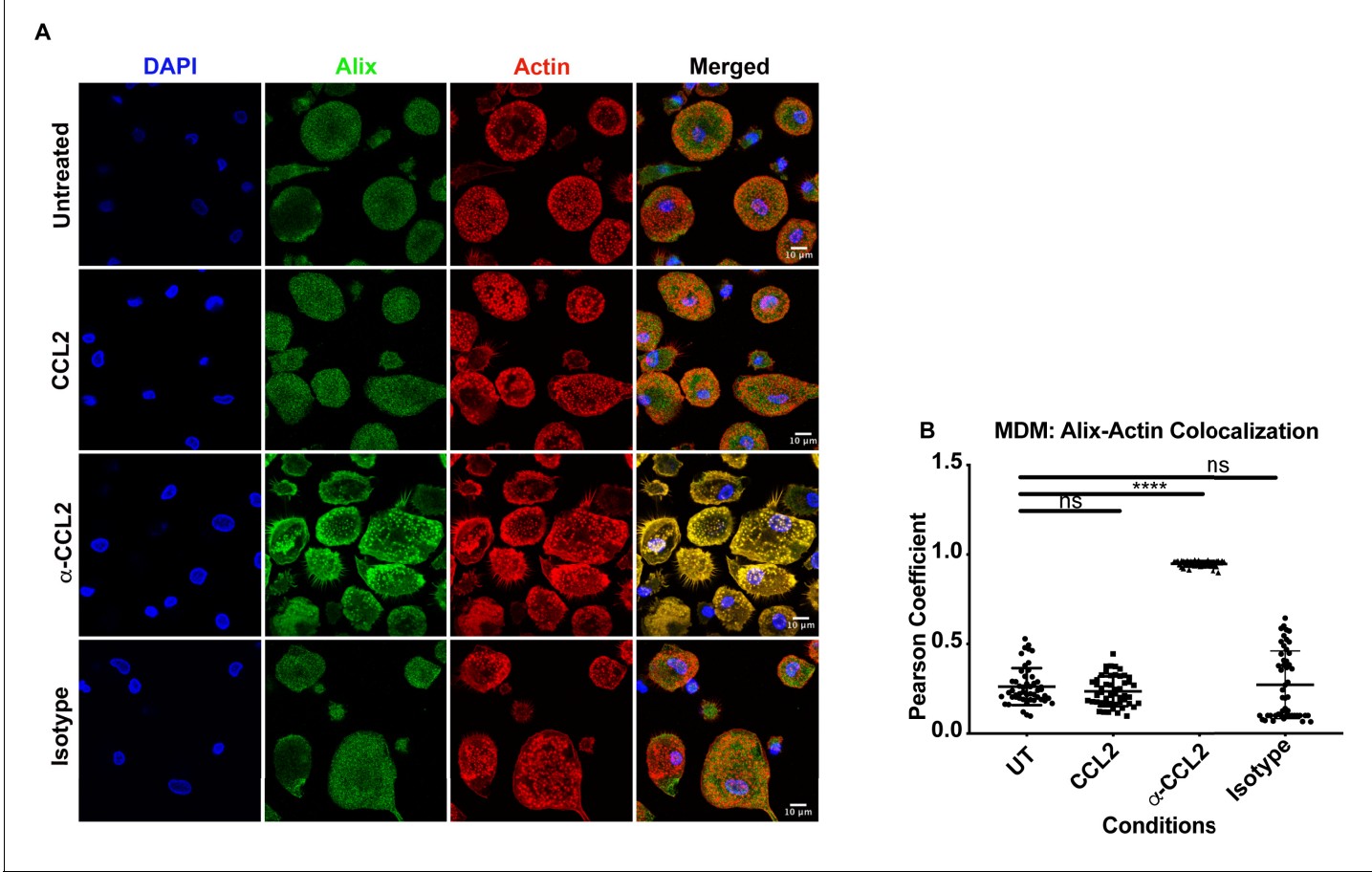

**Figure 7.** CCL2 triggers intracellular redistribution of ALIX in macrophages. (**A**) MDM cells treated similarly as in *Figure 6*, were stained as described above. Fluorescence micrographs show that upon CCL2 depletion, ALIX is redistributed from the cytoplasm to the F-actin structures (podosomes), while in the presence of CCL2 (Untreated and CCL2+) ALIX is diffuse (present as smaller puncta than podosomes) throughout the cytoplasm and near the plasma membrane. (**B**) To quantify the degree of colocalization of ALIX and F-actin under conditions of CCL2 addition and depletion, we calculated the Pearson correlation coefficient between the two fluorophores. For MDMs, the data plotted indicates the close proximity of F-actin and ALIX upon CCL2 depletion, while in the presence of CCL2 (untreated and CCL2) ALIX signal is distinct from that of F-actin. Data are represented as mean ± SEM (n = 100 cells each for MDMs).

DOI: https://doi.org/10.7554/eLife.35546.029

The following source data is available for figure 7:

**Source data 1.** Pearson coefficient values plotted in *Figure 7B*.

DOI: https://doi.org/10.7554/eLife.35546.030

*Kalpana, 2016*). When we measured Gag pixel intensity from the plasma membrane to the nuclear membrane in HIV-1 producing HeLa cells that were untreated or treated with isotype control antibodies, we observed a typical distribution indicative of accumulation of Gag near the plasma membrane (*Figure 8A*, panels in the 'Untreated' and 'Isotype control' rows). This typical distribution was absent in HIV-1 producing HeLa cells treated with α-CCL2 antibodies (see *Figure 8A* 'α-CCL2' panels and *Figure 8B*), which showed a more even distribution of Gag pixel intensities throughout the cytosol (*Figure 8A and B*). These results suggested that Gag does not traffic to the plasma membrane and instead, is distributed thoughout the cytoplasm in the absence of CCL2 and when ALIX is bound to actin cytoskeleton.

We next employed scanning electron microscopy to detect budding virion particles generated in HeLa cells transfected with HIV-1B$_{ADA}$ and examined untreated, α-CCL2 treated and isotype antibody-treated cells at 24 hr post-transfection, which was also the time point where we have measured all virus production in HeLa cells under varying conditions of CCL2 levels. The untreated and isotype

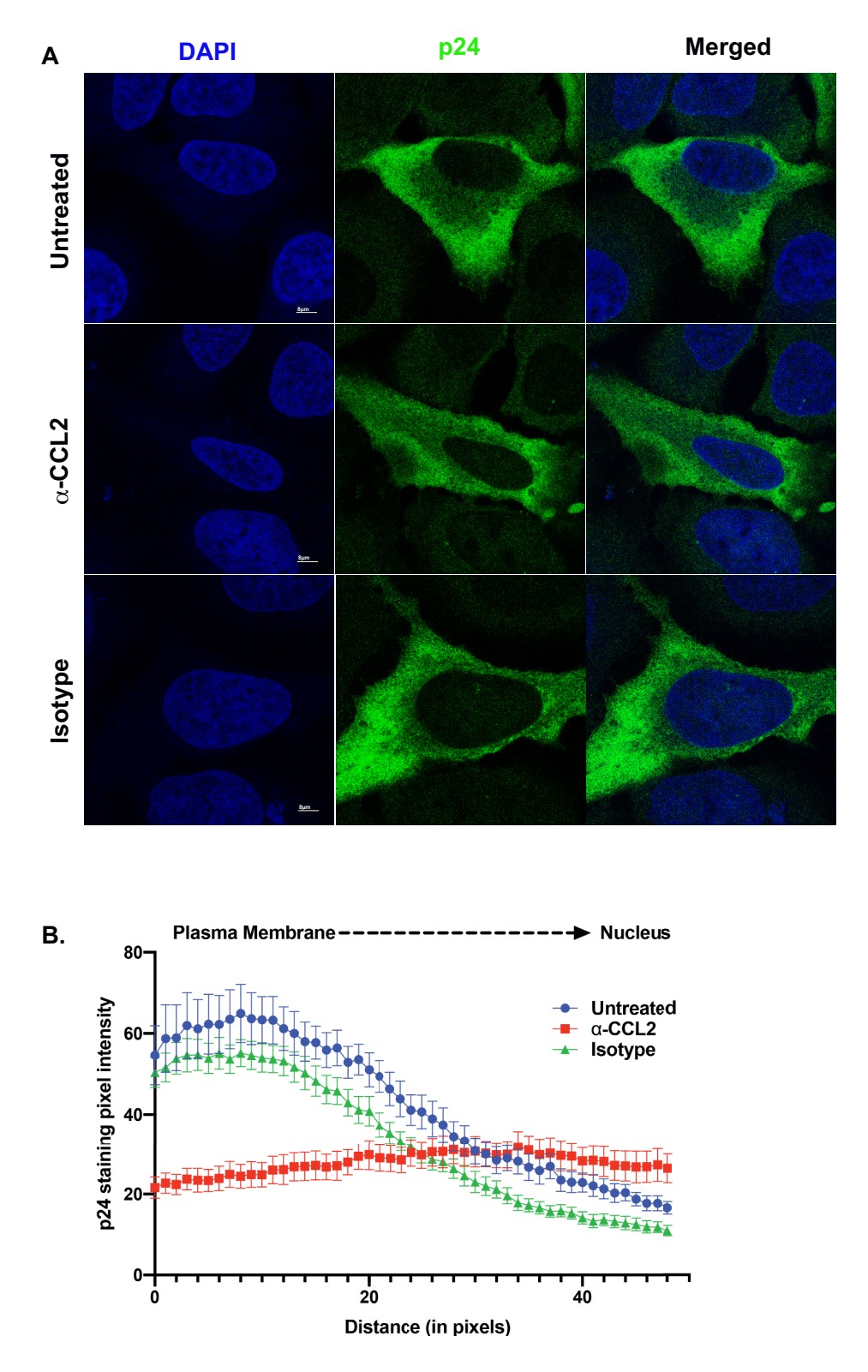

**Figure 8.** ALIX sequestration to actin cytoskeleton alters Gag trafficking. (**A**) Confocal microscopy to visualize HIV-1 p24 in HeLa cells transfected with HIV-1B$_{ADA}$ molecular clone. Cells were treated with 1X PBS, α-CCL2 or Isotype antibody for 24 hr were stained for p24, and DAPI to determine the cellular distribution of p24 upon modulating CCL2 levels. Fluorescence micrographs show that upon CCL2 depletion (α-CCL2 panels), p24 signal is diffused evenly throughout the cytoplasm, while in the untreated and Isotype antibody treated cells, p24 appears to be concentrated near the plasma

*Figure 8 continued on next page*

*Figure 8 continued*
membrane. (B) To quantify the distribution of p24, we quantified p24 pixel intensities starting from plasma membrane to nucleus. Data are represented as mean ± SEM (n = 13) cells.
DOI: https://doi.org/10.7554/eLife.35546.031
The following source data is available for figure 8:

**Source data 1.** Pixel intensity values of p24 staining plotted in *Figure 8B*.
DOI: https://doi.org/10.7554/eLife.35546.032

antibody-treated HeLa cells at 24 hr, as expected, showed production of virions at the cell surface. In the α-CCL2-treated cells, however, only a few virions at the cell surface of a small number of cells (14%) was observed. The majority of cells showed no virus particles on the cell surface but instead displayed virus-like particles in intracytoplasmic vacuoles. In ~35% of the cases, we observed what appear to be virus-like particles (~100 nm) inside vacuoles including those that are still attached to membrane and in early stages of Gag assembly (*Figure 9A*, second row, α-CCL2). In others (~50%), we observed sub-viral-like electron dense particles (~30–40 nm) (*Figure 9A*, third row, α-CCL2) also within intracytoplasmic vacuoles. These results indicated that the unavailability of ALIX leads to perturbation of Gag trafficking and that there is an accumulation of virus-like particles in intracellular vacuoles. While the basis for this unusual phenotype is not clear, ALIX sequestration to actin cytoskeleton appears to have profound effects on HIV-1 Gag trafficking and virion release process.

## HIV-1C is less fit than HIV-1B and the insertion of LY dipeptide in Gag-p6 improves replication fitness

While the data above demonstrate that presence of LY motif is associated with responsiveness to the presence of CCL2, the question remains whether the LYPX motif affects virus fitness when present along with PTAP motif. It has been reported that HIV-1C isolates tend to be less fit in general (*Constantino et al., 2011*). Furthermore, it was reported that Gag-PR of HIV-1C, specifically the absence of L483Y484, is responsible for its low replication capacity (*Kiguoya et al., 2017*). To directly determine the relevance of the presence of LY and the fitness, we compared HIV-1B$_{ADA}$, HIV-1C$_{IndieC1}$ viruses and a mutant of HIV-1$_{IndieC1}$ virus with an LY insertion (HIV-1$_{IndieC+LY}$) that creates the LYPX motif, in a multi-day replication experiment using either PBMCs or MDMs as target cells. Our results showed that HIV-1$_{IndieC1}$ is the least fit and HIV-1B$_{ADA}$ is the most fit of the three viruses and that the insertion of the LY dipeptide in HIV-1C Gag-p6 enhances its replication in both cell types (*Figure 10*). These results were in agreement with the above reports and indicated that LY insertion is responsible for increased fitness.

## Naturally occurring PYxE insertions, like LY insertions, facilitate ALIX-binding and enhance replication via improved virus release of HIV-1C

The results above showed that an LY insertion increased the replication fitness of HIV-1C viruses. Therefore, we wondered if this data has relevance in the natural setting. Variants of HIV-1C that emerged in anti-retroviral therapy failure have been reported in India and they also appear to dominate in Ethiopian HIV-1C isolates in the general HIV-1 infected population (*Neogi et al., 2014*). These insertions were specific to HIV-1C as they were absent in other clades of HIV-1 or circulating recombinant forms (CRFs). These variant viruses contain PYxE tetrapeptide insertions (PYxEi; PYKE, PYRE or PYQE) in place of the missing LY dipeptide in Gag-p6. Since the Y residue in the LYPX motif has been shown to be most critical for ALIX binding to Gag-p6, we hypothesized that the PYxE insertions in HIV-1C may confer the capacity for ALIX-binding, thus increasing overall replication fitness due to increased efficiency of virion release. To test this hypothesis, we selected two of the most prevalent insertions (PYKE and PYRE), generated GST-Gag-p6 fusion proteins of HIV-1C$_{IndieC1}$ with PYRE or PYKE insertions and tested their ability to bind ALIX in a pull-down assay. As expected, GST-HIV-1B p6 bound ALIX but GST-HIV-1C p6 did not (*Figure 11A*). Interestingly, HIV-1C p6 with either PYRE or the PYKE exhibited strong binding to ALIX indicating that these insertions restored HIV-1Cp6-ALIX interaction (*Figure 11A*).

We next tested HIV-1B, HIV-1C, HIV-1C+LY, HIV-1C+PYKE and HIV-1C+PYRE viruses for their replication competence in macrophages in a 30-day replication assay. As our results show, HIV-1C

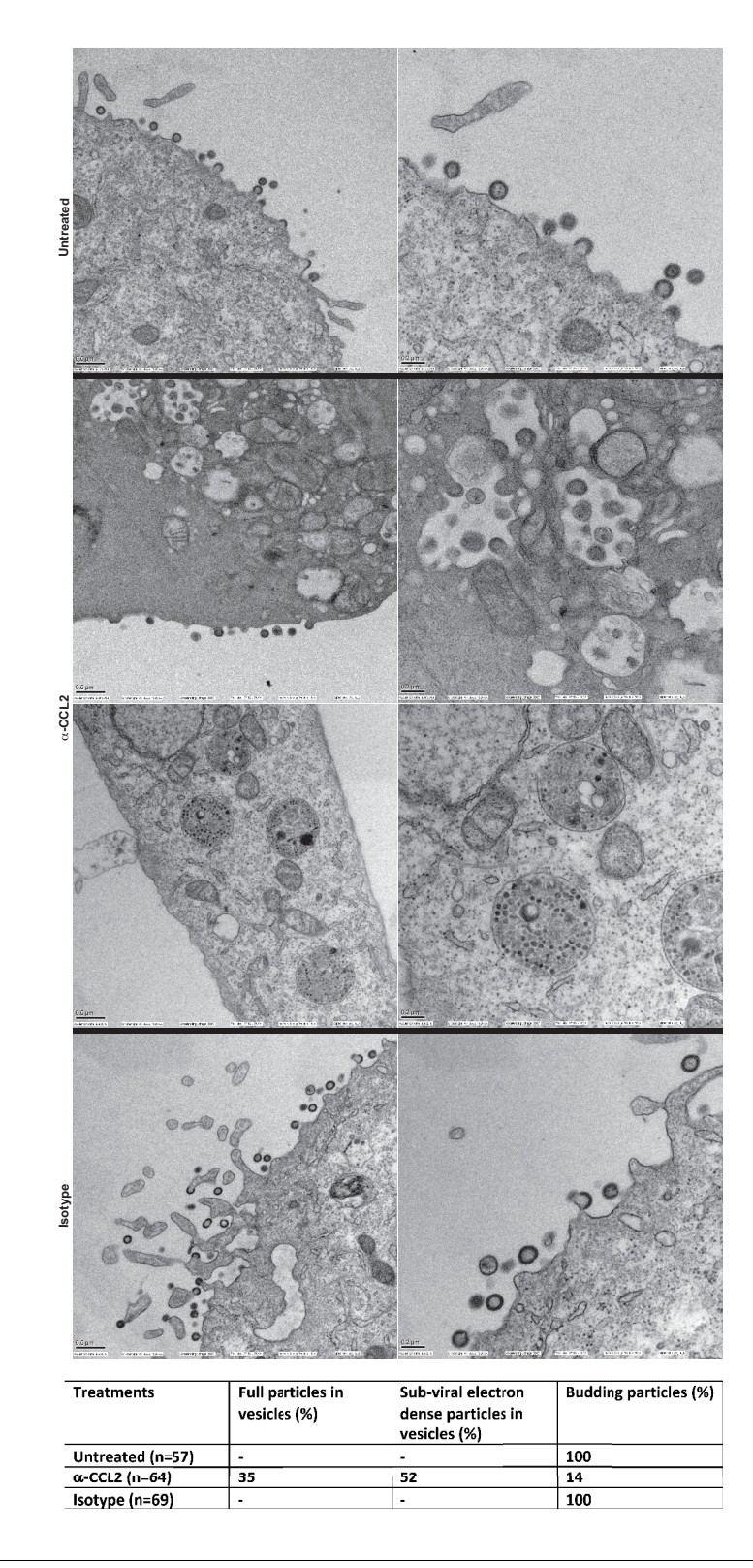

| Treatments | Full particles in vesicles (%) | Sub-viral electron dense particles in vesicles (%) | Budding particles (%) |
|---|---|---|---|
| Untreated (n=57) | - | - | 100 |
| α-CCL2 (n=64) | 35 | 52 | 14 |
| Isotype (n=69) | - | - | 100 |

**Figure 9.** Intracellular compartments of HIV-1 assembly are observed when ALIX is sequestered to actin. (**A**) Electron micrographs to detect HIV-1 budding particles from HeLa cells producing HIV-1B virus particles. HeLa cells tranfected with HIV-1B$_{ADA}$ molecular clone DNA were treated with: nothing (Untreated), α-CCL2 (2.5 μg/mL) or isotype antibody (2.5 μg/mL) for 24 hr post-transfection. The cells were fixed and processed for scanning electron microscopy and imaged at 15,000x, 20,000x and 30,000x magnification. (**B**) The types of virion budding observed was analyzed from 57

*Figure 9 continued on next page*

Figure 9 continued

(Untreated), 64 (α-CCL2-treated) or 69 (Isotype antibody-treated) cells and the results tabulated. Anti-CCL2 treated cell images collected show low level budding at the cell surface, but display more robust intracellular HIV-1 particle and sub-viral particles in cytoplasmic vacuoles.
DOI: https://doi.org/10.7554/eLife.35546.033

was the least replication competent virus, while HIV-1B was the most replication competent (*Figure 11B*). All three insertions led to improved replication capacity, and generally conferred a greater replication capacity when compared to HIV-1C wild type (*Figure 11B*). Similar results were obtained when this same panel of viruses was tested in PBMCs (data not shown). Our results are consistent with reports showing that PYxEi HIV-1C viruses display increased replication rates compared to HIV-1C wild-type virus (*Aralaguppe et al., 2017*; *Chaturbhuj et al., 2018*).

To examine whether the LYPX or PYxE insertions specifically increase replication by enhancing release, we examined the relative virus release properties of HIV-1B$_{ADA}$, HIV-1C$_{IndieC1}$, HIV-1C$_{IndieC1}$ $_{+LY}$, HIV-1C$_{IndieC1+PYKE}$ and HIV-1C$_{IndieC1+PYRE}$ by transfecting each infectious molecular clone into HeLa cells, pelleting the virus in the supernatants through a sucrose cushion and comparing the levels of p24 in the cell lysates and pelleted virus particles. As shown in *Figure 11C*, HIV-1B displayed a higher degree of virus release (greater levels of Gag (p24) in the sucrose cushion pelleted virus preparation than in the cell lysate (pr55)), compared to HIV-1C (lower levels of Gag (p24) in the sucrose cushion pelleted virus preparation and an increased level of Gag (pr55) in the cell lysate). Interestingly, this deficit was corrected with the insertion of LY, PYKE or PYRE into HIV-1C (*Figure 11C*). We conclude that the improved replication competence of HIV-1C PYxE insertion variants is due to improved virion release owing to the acquisition of ALIX-binding capacity.

## Improved replication of PYxEi viruses is ALIX-dependent

To further confirm that the improved replication of PYxEi viruses is indeed mediated by binding of ALIX to p6 and not due to indirect effects, we tested the ability of PYxEi viruses to replicate in the absence of ALIX. We silenced *ALIX* in CEM-CCR5 cells using two separate shRNAs 29394 or 29395 (*Eekels et al., 2011*) and a scrambled shRNA as control. We verified that both shRNAs severely reduced ALIX protein levels (*Figure 12A*). We infected *ALIX* knockdown CEM-CCR5 cells with HIV-1B$_{ADA}$, HIV-1C$_{IndieC1}$, HIV-1C$_{IndieC1}$$^{+LY}$, HIV-1C$_{IndieC1}$$^{+PYRE}$ and HIV-1C$_{IndieC1}$$^{+PYKE}$ and monitored virus production at 18 days post infection. The replication of all viruses with the ability to bind ALIX, namely HIV-1B$_{ADA}$, HIV-1C$_{IndieC1}$$^{+LY}$, HIV-1C$_{IndieC1}$$^{+PYRE}$ and HIV-1C$_{IndieC1}$$^{+PYKE}$ viruses, were reduced in the absence of ALIX (*Figure 12B*). However, replication of viruses lacking the LY dipeptide, which are not dependent on ALIX for virus budding (e.g. HIV-1C$_{IndieC1}$ and HIV-1B$_{ADA-ΔLY}$), were unaffected by *ALIX* silencing – although their replication was about 20–25% lower than HIV-1B$_{ADA}$. These results confirmed that the PYxE insertion not only facilitated ALIX-binding, but also increased virus replication via enhanced particle release that was made possible by the presence of a functional second late motif or its equivalent.

In addition to *ALIX* knock down, we also substantiated the link between the ability to bind ALIX and the efficiency of virus production, by testing the effect of a dominant negative ALIX fragment (ALIX 364–716- V domain; ALIX-DN; *Munshi et al., 2007*). This DN ALIX fragment or wild type ALIX (ALIX WT) were overexpressed in the producer cells and their effect on production of viruses with or without the LYPX or PYxE motif was determined. We found that viruses with an intact LYPX motif or the PYxE insertion were inhibited by the expression of ALIX-DN. In contrast, overexpression of wild-type ALIX stimulated virus production in viruses with LYPX or PYxE motif. Notably, viruses without an LY such as HIV-1C$_{IndieC1}$, were unaffected by either the ALIX-DN or ALIX-WT expression (*Figure 12C*). The intracellular Gag levels were comparable between HeLa cells and cells expressing ALIX-DN However, in cells expressing ALIX-WT, intracellular Gag levels were higher mainly due to increased accumulation of p24 (*Figure 12—figure supplement 1*). The intracellular accumulation of Gag pr55 that is characteristic of α-CCL2 treatment or increase in p24 that is typically observed when CCL2 is added was not observed upon the overexpression of ALIX-DN or ALIX-WT respectively.

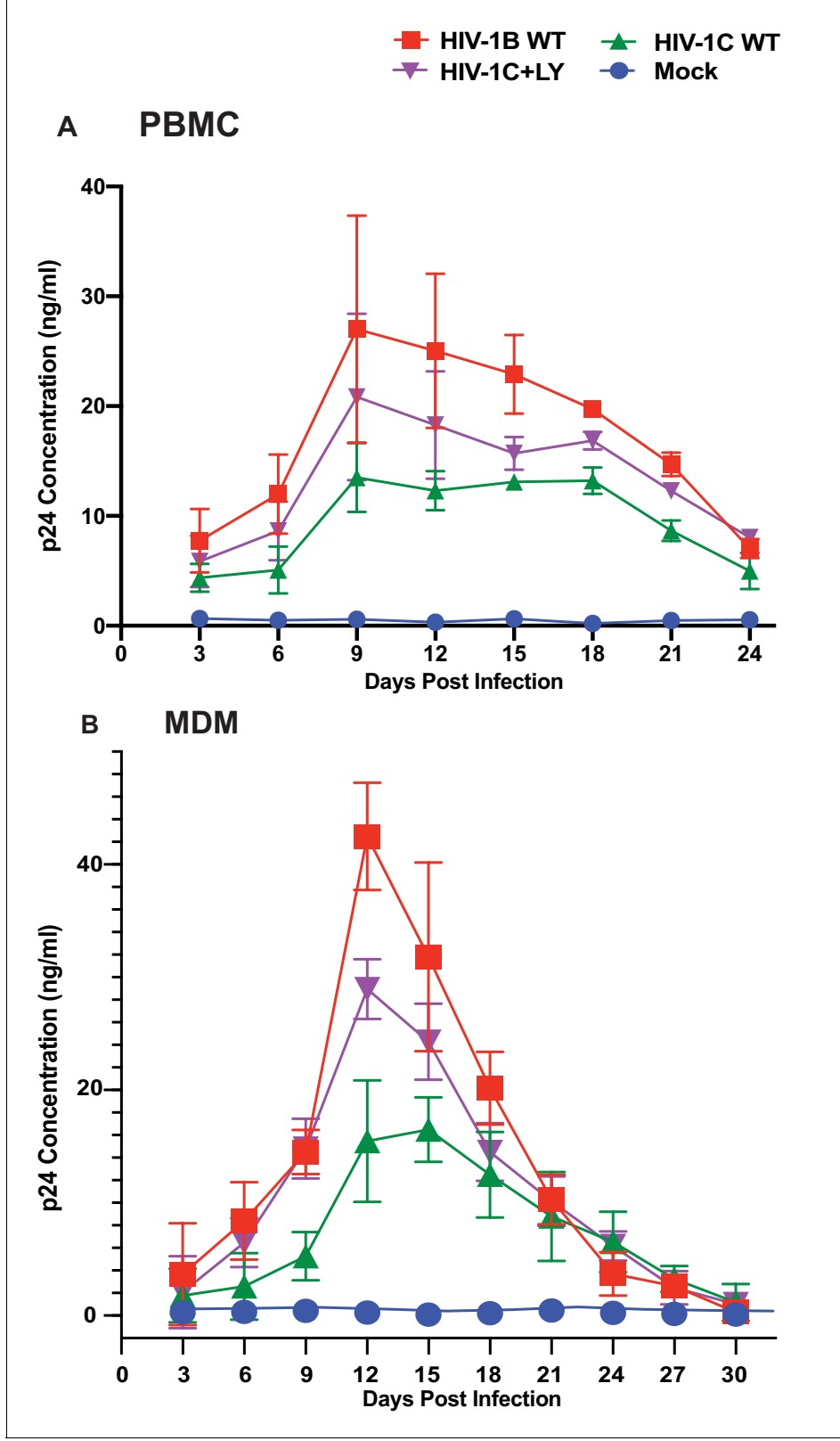

**Figure 10.** Effect of restoring the LYPX motif on HIV-1C replication. Activated PBMCs (Panel A) or MDMs (Panel B) were infected with equal inputs of HIV-1B$_{ADA}$, HIV-1C$_{IndieC1}$ and HIV-1C$_{IndieC1}$+LY viruses. Aliquots of media were collected every 3 days until day 30 and the p24 levels were determined (PBMCs, n = 2 and macrophage, n = 3). Data are represented as mean ± SEM. Results indicate increased replication capacity of LYPX-containing viruses.

*Figure 10 continued on next page*

*Figure 10 continued*

DOI: https://doi.org/10.7554/eLife.35546.034
The following source data is available for figure 10:
**Source data 1.** All p24 values plotted in *Figure 10A*.
DOI: https://doi.org/10.7554/eLife.35546.035
**Source data 2.** All p24 values plotted in *Figure 10B*.
DOI: https://doi.org/10.7554/eLife.35546.036

## PYxE insertions also confer CCL2-responsiveness to HIV-1C

Finally, we tested whether HIV-1C viruses carrying these insertions were responsive to CCL2. We have shown that HIV-1C, due to its inability to bind ALIX, is insensitive to the presence or absence of CCL2 in the media. To evaluate if PYxE insertion confers CCL2 responsiveness, we transfected molecular clones of both HIV-1$_{IndieC1}$ PYxEi viruses into HeLa cells and examined the effect of adding CCL2 or α-CCL2 antibodies to the medium on virus production. We found that, unlike the parent HIV-1C viruses lacking LY motif, PYxEi viruses had regained CCL2-responsiveness (*Figure 13*). Control viruses with deletion of LY in HIV-1B$_{ADA}$ were unresponsive to CCL2, while LY insertion in HIV-1C restored CCL2 responsiveness (*Figure 13*). These results together indicate that the presence of LY or PYxE insertions improved virus replication due to improved particle release due to restoration of ALIX binding and thus responsiveness to CCL2.

## Discussion

In this report, we have described a previously unknown role for the β-chemokine CCL2 in mobilizing ALIX, an ESCRT adaptor protein and enhancing viral particle production. We demonstrate that CCL2 increases the availability of cytosolic ALIX by triggering its dissociation from the actin cytoskeleton, facilitating binding to Gag-p6 and thus enhancing HIV-1 particle release and replication. We also demonstrate that CCL2 regulation of HIV-1 replication is clade-specific and is manifested by genetic differences at the second late motif, LYPX. HIV-1B viruses contain an LYPX motif, in addition to PTAP motif, and their virion release is increased in the presence of CCL2 due to greater access to ALIX. In contrast, due to the absence of LYPX motif, HIV-1C cannot utilize ALIX irrespective of the degree of its availability, thus its replication cannot be further enhanced. Interestingly, despite the lack of LYPX motif, certain HIV-1C isolates (such as those emerging in drug failure cases in India or present in greater than 50% of all Ethiopian isolates tested [*Neogi et al., 2014*]) have acquired PYxE insertions in Gag-p6, which confer ALIX-binding, CCL2-mediated increase in virus release and improved replication fitness. All these data are in support of a role for CCL2 chemokine in facilitating HIV-1 replication by mobilizing ALIX.

CCL2 is a chemokine elaborated by cells infected by many different pathogens to attract monocytes as an innate immune mechanism. Our results suggest a novel means by which HIV-1 viruses that can bind ALIX can exploit the increased ALIX accessibility, as a result of increased CCL2 in their micro-environment, to gain a competitive edge. Both CCL2 and its receptor, CCR2, are widely expressed in most HIV-susceptible cell types, although CD4 T cells appear to lack expression of CCL2. Nevertheless, the expression of CCR2 on both macrophages and CD4 T cells indicate that in vivo conditions that lead to increased CCL2 production are beneficial for replication of HIV strains harboring a Gag capable of binding ALIX.

It was surprising to find virus-like particles trapped in intracytoplasmic vacuoles in HeLa cells (in the presence of α-CCL2), rather than at the plasma membrane, as was observed when a dominant negative ALIX form was over-expressed in HIV producer cells (*Munshi et al., 2007*). It is already well established that plasma membrane is the primary site of virus assembly and release (*Jouvenet et al., 2006*). Therefore, this observation - especially in HeLa cells - is difficult to reconcile with. However, our experiment is different from expressing a dominant negative ALIX (where some endogenous wild type ALIX is still present) in that nearly all ALIX can be completely trapped in actin cytoskeleton when CCL2 is immunodepleted. Therefore, further investigation is required to understand the basis for intravacuolar virus-like particles observed in response to CCL2 immuno-depletion our experiments.

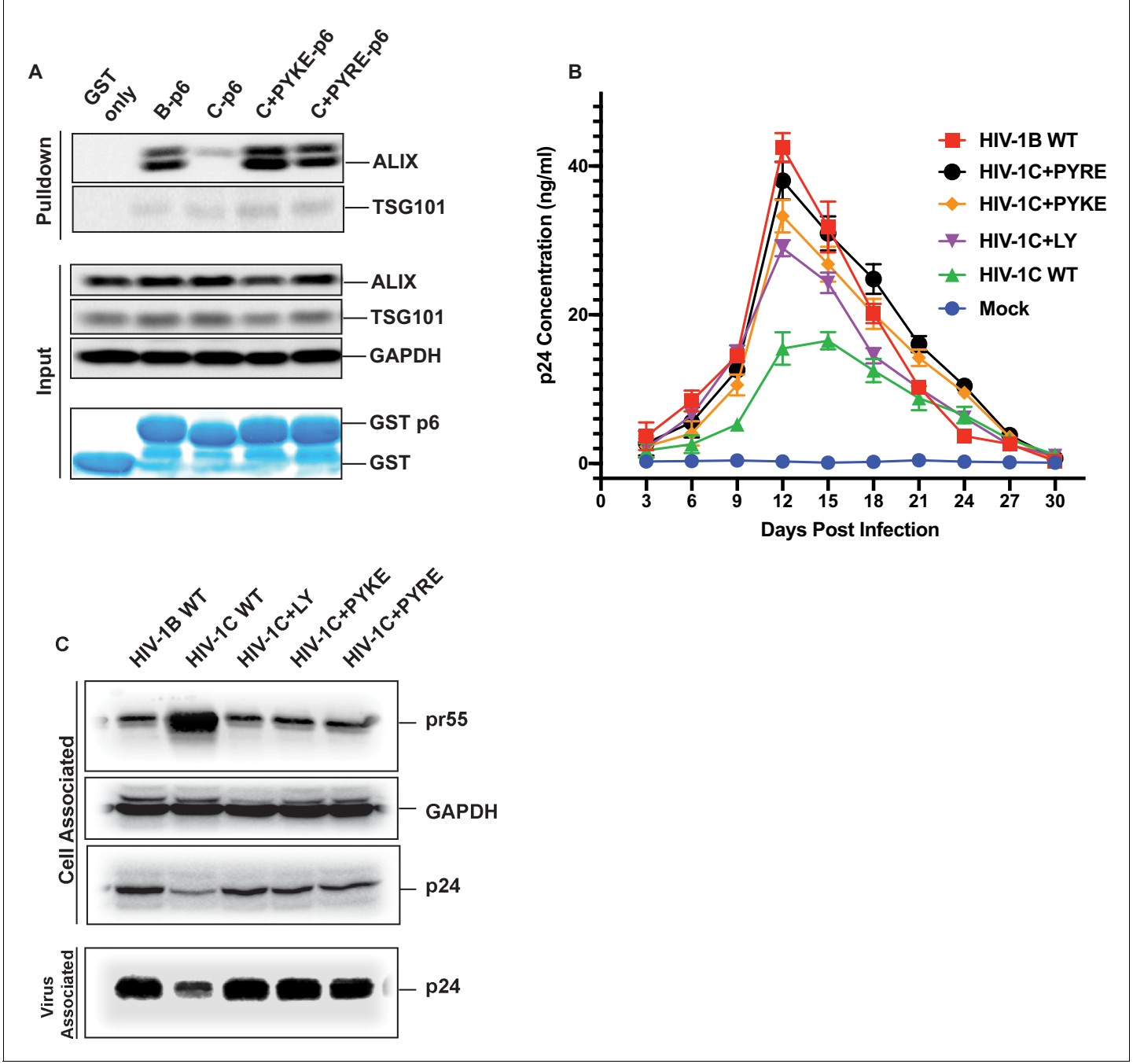

**Figure 11.** PYxE insertions restore ALIX binding to Gag-p6 and enhance replication capacity due to improved virion release. (A) PYxE insertions restore ALIX binding to HIV-1C Gag-p6. GST protein, either by itself or fused to HIV-1B-p6, HIV-1C-p6, HIV-1C+PYxE-p6 were expressed and purified from *E. coli* using Glutathione agarose beads, and incubated with HeLa cell lysates. Pull-downs of the protein complexes were resolved on SDS-PAGE gels and the western immunoblots probed for ALIX and TSG101 (*top panel*). Input levels of ALIX, TSG101 as well as the GAPDH control are in the *middle panels* and purified GST-only and GST-p6 fusion inputs are in *lower panels*. The experiment was performed three times. (B) The effect of PYxE insertion on virus replication fitness. MDMs were infected with equal p24 and media were collected every 3 days till day 30. Media collected were clarified and p24 levels determined via p24 ELISA. Data are represented as mean ± SEM. The experiment was performed three times. (C) The effect of PYxE insertion on virus release. HeLa cells were transfected with molecular clones. Media were collected 24 hours post-transfection and virus pelleted through a 20% sucrose cushion. Cells were pelleted and lysed. Both virus and cell lysates were subjected to a western immunoblot followed by detecting virus-associated p24 and cell-associated p24 and other Gag-derived proteins (pr55, (**p41**) with α-p24 antibodies. GAPDH was used as a loading control.
DOI: https://doi.org/10.7554/eLife.35546.037

The following source data is available for figure 11:

*Figure 11 continued on next page*

*Figure 11 continued*

**Source data 1.** All p24 values plotted in *Figure 11B*.
DOI: https://doi.org/10.7554/eLife.35546.038

Our data suggests that the availability of robust levels of ALIX, and in turn a significant level of ALIX-p6 interaction to increase virion budding, is modulated by the levels of CCL2, which is required to mobilize ALIX from F-actin. In fact, it is not unlikely that the low level of HIV-1 production observed in the presence of α-CCL2 is due to the residual levels of CCL2 or a form of ALIX that may still be available. Determining whether CCL2 is obligatory for HIV budding necessitates generating CRISPR knockout cell lines of *CCL2* and/or its receptor, *CCR2*. Other retroviruses, such as Equine Infectious Anemia Virus (EIAV), Rous Sarcoma Virus (RSV) and Murine Leukemia Virus (MuLV)) as well as RNA viruses (Sendai virus and Yellow fever virus) and DNA viruses (Vaccinia) also employ a LYPX motif (*Votteler and Sundquist, 2013*) for budding. Many of these viruses are known to induce CCL2, suggesting that the exploitation of CCL2-induced ALIX for viral replication may be a feature shared across diverse virus orders.

We find that in all cells employed here except CD4 T cells, the stimulation of virus release by the addition of CCL2 is much smaller than the level of inhibition observed upon CCL2 depletion. We believe that this asymmetry reflects the amount of ALIX that is available in the actin cytoskeletal compartment for release by CCL2 signaling. On the other hand, depletion of CCL2 can direct most of the ALIX to actin cytoskeleton bringing about maximal inhibition of virus production. It is likely that in CD4 T cells, which hardly produce any CCL2, a lack of signaling can result in most of the ALIX being already associated with actin at steady state resulting in a smaller degree of inhibition upon CCL2 depletion.

We selected CCL2 as our focus, as HIV-1B infection of macrophages induces CCL2 levels and because plasma levels of CCL2 correlate with viremia. HIV-1 may have evolved to benefit from the increasing CCL2 in its micro-environment. Numerous studies have linked CCL2 but not other β-chemokines to increase in virus replication. Nevertheless, there is a possibility that other β-chemokines may increase ALIX mobilization from the cytoskeleton. These are important future avenues of investigation.

The precise mechanism by which CCL2 signaling leads to the release of ALIX from the cytoskeleton is unknown and needs further investigation. It is known that CCL2 binding to CCR2 modulates a variety of cellular pathways – most of them poorly understood (*Covino et al., 2016*). CCL2 binding of CCR2 signals through Frount, which eventually leads to monocyte migration (*Terashima et al., 2005*). It remains to be investigated whether Frount plays any role in the effects we report here. Both CCL2 signaling (*Lee et al., 2009*) and ALIX (*Cabezas et al., 2005*; *Pan et al., 2006*) (*Chu et al., 2012*) are known to be involved in actin reorganization. In addition, ALIX is known to interact with F-actin structures as well as actin-binding proteins – for example, cortactin or α-actinin (*Pan et al., 2006*). Thus, ALIX-F-actin interaction may also be modulated by many factors including ALIX-binding proteins (e.g. ALG-2) or post-translational modifications of ALIX, actin or actin-binding proteins. Further investigation is required to delineate the specific pathway that mediates CCL2-mediated ALIX mobilization from actin.

One intriguing finding from our data is that preventing HIV-1B Gag-p6-ALIX association by blocking access to ALIX (via CCL2 depletion) inhibited virus production from wild type HIV-1B (with an intact PTAP and LYPX domains) to the same degree as HIV-1B$_{ADA-PTAP}$ mutant in the absence of any treatment (*Figure 4B*, compare HIV-1B wild type, α-CCL2-treated lane to PTAP-, Untreated lanes). In other words, blocking access to ALIX can bring about as much inhibition of HIV-1B virus production as does mutating PTAP motif. This may be due to the unique role played by ALIX in PTAP-mediated ESCRT III recruitment. Cellular processes that require ESCRT complexes are known to serially recruit ESCRT 0, ESCRT I, ESCRT II and ESCRT III through protein-protein interactions intrinsic to ESCRT complexes (ESCRT 0 recruits ESCRT I, ESCRT I recruits ESCRT II, ESCRT II recruits ESCRT III and ESCRT III recruitment leads to the recruitment of VPS4 AAA ATPase and associated proteins) (*McCullough et al., 2018*). In the case of HIV-1B budding, although the recruitment of ESCRT I and

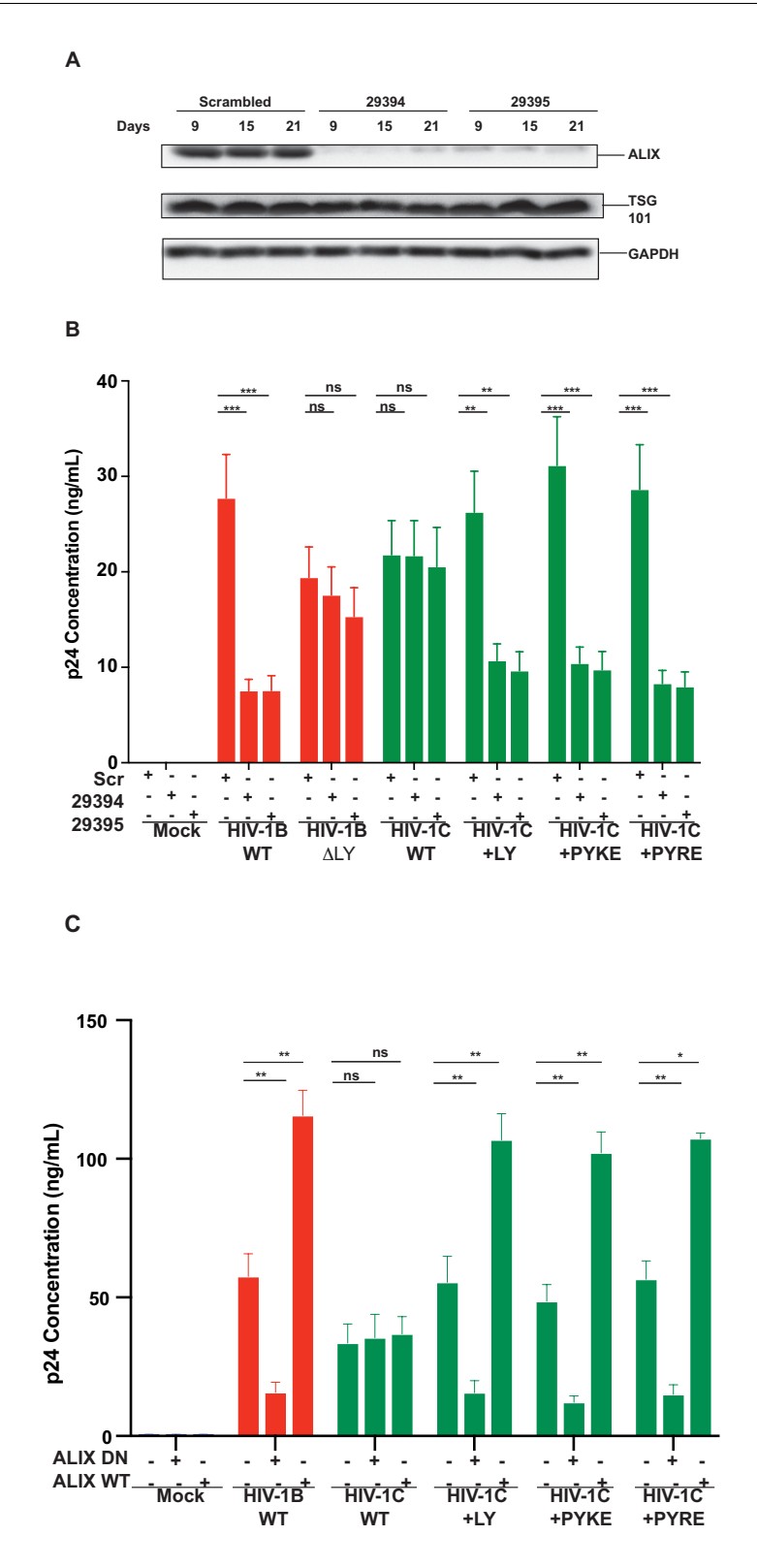

**Figure 12.** Replication of viruses bearing LYPX or PYxE insertions is inhibited by silencing ALIX or by ALIX dominant negative protein but enhanced by ALIX over expression. (**A**) CEM-CCR5 cells were transduced with scrambled shRNA or anti-ALIX shRNA (29394 or 29395) expression cassettes. After selecting for transduction, the presence of ALIX protein was determined by western immunoblotting. (**B**) The shRNA-transduced cells were infected with HIV-1B, HIV-1B(ΔLY), HIV-1C, HIV-1C+LY, HIV-1C+PYKE or HIV-1C+PYRE viruses. Media were collected every 3 days and p24 levels for day 18 at

*Figure 12 continued on next page*

*Figure 12 continued*

the peak virus replication determined. Data are represented as mean ± SEM (n = 3). (**C**) HeLa cells were co-transfected with indicated HIV-1 molecular clones and ALIX wild type or ALIX V domain (ALIX-DN) expression plasmids. At 24 hr post-transfection, p24 levels were measured. Data are represented as mean ± SEM (n = 3).

DOI: https://doi.org/10.7554/eLife.35546.039

The following source data and figure supplement are available for figure 12:

**Source data 1.** All p24 values plotted in *Figure 12B*.

DOI: https://doi.org/10.7554/eLife.35546.041

**Source data 2.** All p24 values plotted in *Figure 12C*.

DOI: https://doi.org/10.7554/eLife.35546.042

**Figure supplement 1.** Effect of over-expression of wild type and dominant negative ALIX on intracellular capsid protein.

DOI: https://doi.org/10.7554/eLife.35546.040

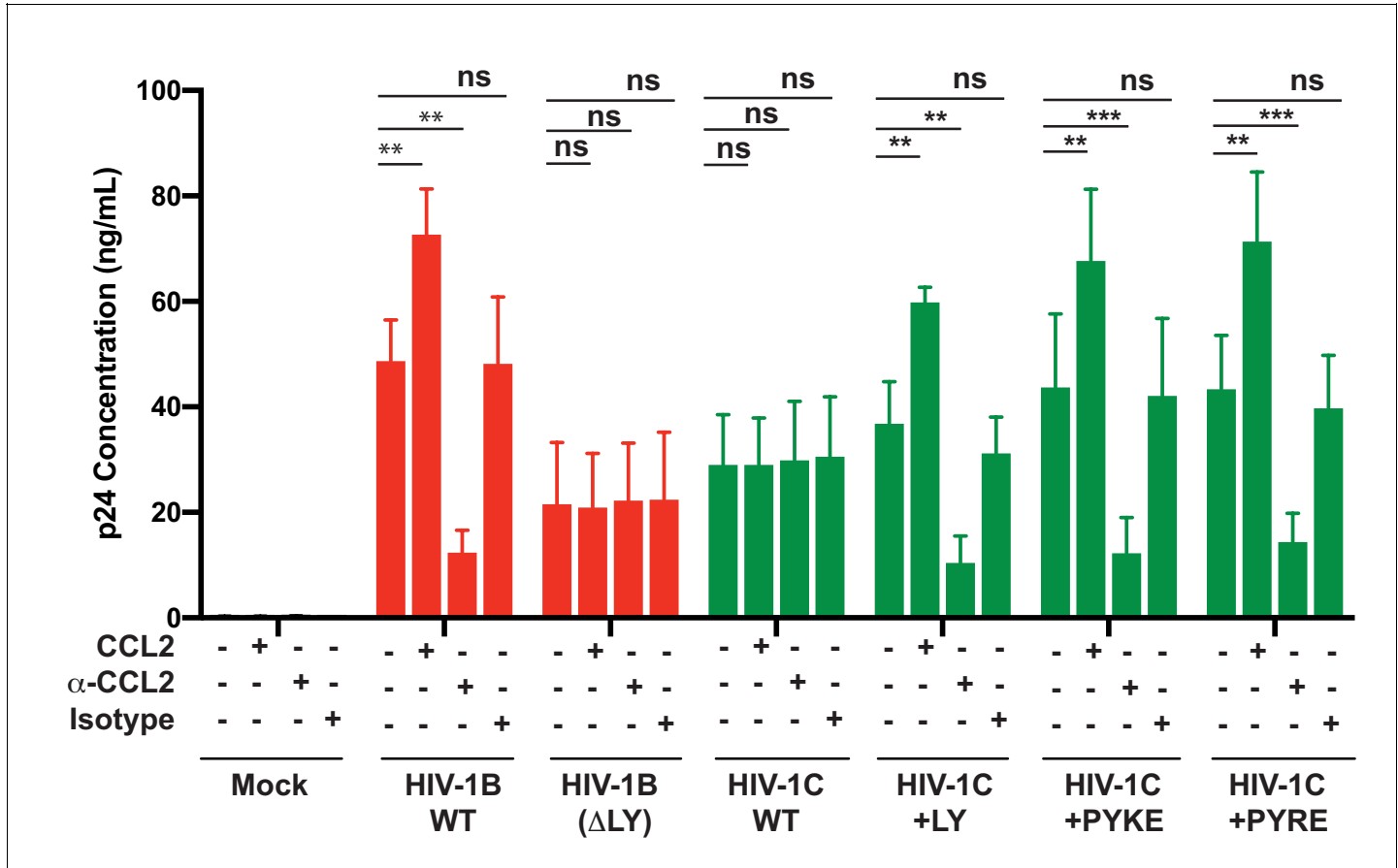

**Figure 13.** PYxE insertions restore CCL2-mediated increase in virus production in HIV. HeLa cells were transfected with equal DNA concentrations of HIV-1B$_{ADA}$, HIV-1C$_{IndieC1}$ and HIV-1C$_{IndieC1}$+LY or HIV-1C$_{IndieC1}$+PYxE molecular clones. Transfected cells were treated with nothing (untreated), CCL2, α-CCL2 or with isotype antibodies. After 24 hr post-transfection, media were collected, clarified and p24 levels determined. Data are represented as mean ± SEM. The experiment was performed three times.

DOI: https://doi.org/10.7554/eLife.35546.043

The following source data is available for figure 13:

**Source data 1.** All p24 values plotted in *Figure 13*.

DOI: https://doi.org/10.7554/eLife.35546.044

ESCRT III are accomplished independently via PTAP-TSG101-ESCRT I axis or LYPX-ALIX-ESCRT III axis, respectively, the TSG101-ESCRT complex can recruit ESCRT III and initiate budding without the need for ESCRT II complex (*Langelier et al., 2006*). As HIV-1B employs two parallel means of recruiting ALIX into the virion budding sites (via p6-LYPX or NC-Bro domain), ALIX could serve as a means of bridging ESCRT I and ESCRT III in PTAP-mediated HIV-1 budding. The complete sequestration of ALIX in actin cytoskeleton removes two major ways of recruiting ESCRT III leaving only the minor pathways of either ESCRT II-CHMP6 or Vps28-CHMP6 to recruit ESCRT III complex. A similar mechanism may explain stronger inhibition of HIV-1B virus budding in the presence of dominant negative ALIX protein when compared to CCL2 depletion (*Figure 12*; *Munshi et al., 2007*).

The effect of CCL2 immuno-depletion in MDMs on reducing HIV-1B replication has been previously shown (*Fantuzzi et al., 2003*). A more recent report (*Sabbatucci et al., 2015*) suggested that CCL2 depletion in HIV-infected MDMs leads to impairment of viral DNA accumulation and integration. The mechanism for this effect was proposed to be due to the induction of APOBEC3A suggesting that the CCL2-depletion affects an early event. Our results demonstrate that CCL2-mediated effects are on the late events and that the mere insertion of an LY dipeptide to HIV-1C could confer CCL2-responsiveness and its deletion from HIV-1B could abrogate it. Furthermore, in single-cycle infectivity tests using TZMbl cells, we did not see any effect of CCL2 depletion on the establishment of infection (*Figure 4—figure supplement 2*). Thus, we believe that the results we have reported are mainly due to the effect of CCL2 on late event where ALIX redistribution leads to virion release.

It is intriguing that most clades of HIV-1, with the exception of clade C, display high conservation of a functional Gag-p6 LYPX motif, which facilitates binding to ALIX (*Patil and Bhattacharya, 2012*) and through an undefined determinant, an ability to induce CCL2, which in turn, increases cytoplasmic availability of ALIX for Gag-p6. Clade C viruses, which are known to have the lowest pathogenic fitness in group M HIV-1 (*Ariën et al., 2007*), exhibit a conserved absence of the LYPX motif and inability to induce robust CCL2. A general reduced replication fitness of HIV-1 may be an evolutionary tool to replicate at low levels, evading or delaying recognition by host immune surveillance, ensuring its survival in the host for a longer time and facilitating transmission to more individuals. The absence of LYPX and the relative inefficiency with which the envelope protein facilitates HIV-1C entry both serve this purpose. Reasons for the success of HIV-1C as the most prevalent clade may be a combination of its well-conserved, reduced low fitness along with an ability to emerge as higher fitness variants under certain circumstances. Our data indeed shows that the PYxE insertions enhance virion release (*Figure 11C*). PYxE insertions have been reported exclusively in HIV-1C isolates and not in HIV-1B suggesting that the acquisition of PYxEi *via* recombination is facilitated by a selective advantage conferred on a virus displaying sub-optimal budding. It has been suggested that HIV-1C viruses in India show an evolutionary trend toward PTAP-duplications from 1990s to 2015 (~15% to 30%) (*Sharma et al., 2018*). Along the same lines, it has been reported that HIV-1C viruses in India, which normally have three NFκB sites in the LTR, evolve into a more fit virus by acquiring a fourth NFκB site (*Bachu et al., 2012*) that allows the virus to replicate better due to an overall increase in basal transcription. In fact, the study reports that there is a tendency for HIV-1C to evolve toward 4 NFκB sites. These three different means of HIV-1C evolution into a more fit virus by acquiring duplicate PTAP motif, NFκB site or PYxE insertions suggest that HIV-1C, due to its low replication fitness, is evolving approaches to achieve greater fitness to survive in a patient.

ALIX is involved in multiple cellular functions. It binds to the ESCRT III component CHMP4 (*Martin-Serrano et al., 2003a*; *Martin-Serrano et al., 2003b*) to facilitate multivesicular body formation; it binds Cep55 to facilitate cytokinesis (*Carlton et al., 2008*); and it binds to actin and promotes cytoskeletal assembly in fibroblasts and HeLa cells (*Cross and Woodroofe, 1999*; *Pan et al., 2006*). Although ALIX association with the cytoskeletal network and its presence in the lamellipodia are well known (*Pan et al., 2006*), its association and dissociation from the actin fibers in response to CCL2 is a novel finding of the current report. This finding is likely to shed novel insights into both mechanisms of HIV-1 virus production and the function of these proteins in basic biological functions such as cytokinesis. The role of CCL2 has been of interest in cancer biology (*Loberg et al., 2006*; *Lu et al., 2006*; *Roca et al., 2008*; *Sato et al., 1995*) and the discovery of CCL2-mediated redistribution of ALIX can potentially have wide-ranging implications.

# Materials and methods

## Key resources table

| Reagent type (species) or resource | Designation | Source or reference | Identifiers | Additional information |
|---|---|---|---|---|
| Cell line (*H. sapiens*) | HeLa | ATCC | RRID:CVCL_0030 | |
| Cell line (*H. sapiens*) | 293T | ATCC | CRL-3216 | |
| Cell line (*H. sapiens*) | TZMbl | NIH AIDS Reagent repository | 8129 | |
| Cell line (*H. sapiens*) | U87-CCR2 (U87 CD4+ CCR2+ cells) | NIH AIDS Reagent repository | 4033 | |
| Cell line (*H. sapiens*) | CEM.NK$^R$ CCR5 (CEM-CCR5) cells | NIH AIDS Reagent repository | 4376 | |
| Recombinant DNA reagent | Molecular clone HIV-1$_{ADA}$ (HIV-1B) | *Theodore et al., 1996* | | Dr. Keith Peden |
| Recombinant DNA reagent | Molecular clone HIV-1$_{IndieC1}$ (HIV-1C) | *Mochizuki et al., 1999* | | Dr. Udaykumar Ranga, Jawaharalal Nehru Centre for Advanced Research, Bangalore, India |
| Recombinant DNA reagent | GFP expression plasmid peGFP n1 | Clontech | 6085–1 | For eukaryotic expression under CMV promoter. Includes Kanamycin resistance/ Neomycin resistance marker |
| Recombinant DNA reagent | MISSION Lentiviral Packaging mix pLKO | Sigma Aldrich PMID: 16929317 | SHP001 | Includes two plasmids: pMission Gag-Pol and pCMV VSV-G both - carrying ampicillin resistance marker |
| Recombinant DNA reagent | pLKO.1 construct expressing shRNA targeting *ALIX* | Sigma Aldrich *Eekels et al., 2011*; PMID: 16929317 | Clone ID TRCN0000029394 | Target sequence: 5' GCTGCTAAACATTACC AGTTT 3' Carries the puromycin-resistance marker |
| Recombinant DNA reagent | pLKO.1 construct expressing shRNA targeting *ALIX* | Sigma Aldrich *Eekels et al., 2011*; PMID: 16929317 | Clone ID TRCN0000029395 | Target sequence: 5' CCTGAATTACTGC AACGAAAT 3' Carries the puromycin-resistance marker |
| Recombinant DNA reagent | pLKO.1 encoding sequences for scrambled shRNA control | Sigma Aldrich *Eekels et al., 2011*; PMID: 16929317 | SHC001 | Carries puromycin -resistance marker |
| Recombinant DNA reagent | pBluescript-KS | Stratagene | X52331.1 | Bacterial cloning plasmid with a polylinker |
| Recombinant DNA reagent | pSL1190 | Amersham | | Bacterial cloning plasmid with a polylinker |
| Recombinant DNA reagent | HIV-1$_{ADA}$ mutant Gag p6ΔLY | This paper | | Molecular clone of a HIV-1ADA clade B variant created by site directed mutagenesis |
| Recombinant DNA reagent | HIV-1ADA mutant Gag p6ΔPTAP(+LIRL) | This paper | | Molecular clone of a HIV-1ADA clade B variant created by site directed mutagenesis |
| Recombinant DNA reagent | HIV-1ADA mutant Gag p6ΔPTAP(+LIRL) | This paper | | Molecular clone of a HIV-1ADA clade B variant created by site directed mutagenesis |

*Continued on next page*

*Continued*

| Reagent type (species) or resource | Designation | Source or reference | Identifiers | Additional information |
|---|---|---|---|---|
| Recombinant DNA reagent | HIV-1ADA mutant ΔLY | This paper | | Molecular clone of a HIV-1ADA clade B variant created by site directed mutagenesis |
| Recombinant DNA reagent | HIV-1$_{IndieC1}$ mutant Gag p6+LY | This paper | | Molecular clone of a HIV-1IndieC1 clade C variant created by site directed mutagenesis |
| Recombinant DNA reagent | HIV-1$_{IndieC1}$ mutant Gag p6ΔPTAP(+LIRL) | This paper | | Molecular clone of a HIV-1IndieC1 clade C variant created by site directed mutagenesis |
| Recombinant DNA reagent | HIV-1$_{IndieC1}$ mutant Gag p6ΔPTAP(+LIRL) | This paper | | Molecular clone of a HIV-1IndieC1 clade C variant created by site directed mutagenesis |
| Recombinant DNA reagent | HIV-1C$_{IndieC1}$ + PYKE | This paper | | Molecular clone of a HIV-1IndieC1 clade C variant created by site directed mutagenesis |
| Recombinant DNA reagent | HIV-1C$_{IndieC1}$ + PYRE | This paper | | Molecular clone of a HIV-1IndieC1 clade C variant created by site directed mutagenesis |
| Recombinant DNA reagent | pGEX-6Pi | GE Healthcare | 28954648 | For expression of Glutatione S Transferase (GST) protein in *E. coli* |
| Recombinant DNA reagent | pGEX-6Pi-Bp6 | This paper | | For expression of GST-Bp6 fusion protein in *E. coli* |
| Recombinant DNA reagent | pGEX-6Pi-Cp6 | This paper | | For expression of purified GST-Cp6 fusion protein *E. coli.* |
| Recombinant DNA reagent | pGEX-6Pi-C + PYKE p6 | This paper | | For expression of purified GST-C + PYKEp6 fusion protein in *E. coli.* |
| Recombinant DNA reagent | pGEX-6Pi-C + PYRE p6 | This paper | | For expression of purified GST-C + PYREp6 fusion protein in *E. coli.* |
| Antibody | anti-CCL2 (Mouse Monoclonal) | PeproTech | 500-M71-500UG | 2.5 µg/mL |
| Antibody | Isotype antibody IgG2a, kappa (Mouse polyclonal) | PeproTech | 500-M00 | 2.5 µg/mL |
| Antibody | anti-ALIX (mouse Monoclonal) | Thermo-Scientific | MA183977 | (1:500) |
| Antibody | anti-TSG101(mouse Monoclonal) | GeneTex | GTX70255 | (1:500) |
| Antibody | anti-beta-actin (mouse Monoclonal) | Thermo-Scientific | MA515739 | (1:15000) |
| Antibody | anti-GAPDH (mouse Monoclonal) | Thermo-Scientific | MA515738 | 1:10,000 |
| Antibody | anti-HIV p24 (goat) | Gift from Dr. David Otts (NCI) | Gift from Dr. David Otts (NCI) | (1:5000) Dr. David Ott, NCI, Frederick |

*Continued on next page*

*Continued*

| Reagent type (species) or resource | Designation | Source or reference | Identifiers | Additional information |
|---|---|---|---|---|
| Antibody | anti-Goat antibodies Alexa 800 nm (Donkey polyclonal) | Life Technologies | A32930 | 1:10,000 |
| Antibody | anti-mouse antibodies Alexa 680 nm (goat polyclonal) | Life Technologies | A-21058 | 1:10,000 |
| Antibody | anti mouse-Alexa 488 antibody (goat polyclonal) | Life Technologies | A-11001 | 1:250 |
| Antibody | anti-goat Alexa 488 (Donkey polyclonal) | Life Technologies | A-11055 | 1:500 |
| Peptide, recombinant protein | GST | This paper | | Recombinant protein purified using Glutathione agarose beads from lysates of *E. coli* expressing pGEX-6Pi described in this Table |
| Peptide, recombinant protein | GST-HIV-1Bp6 | This paper | | Recombinant protein purified using Glutathione agarose beads from lysates of *E. coli* expressing pGEX-6Pi-Bp6 described in this Table |
| Peptide, recombinant protein | GST-HIV-1Cp6 | This paper | | Recombinant protein purified using Glutathione agarose beads from lysates of *E. coli* expressing pGEX-6Pi-Cp6 described in this Table |
| Peptide, recombinant protein | GST-HIV-1C-PYKE p6 | This paper | | Recombinant protein purified using Glutathione agarose beads from lysates of *E. coli* expressing pGEX-6Pi-C + PYKE p6 described in this Table |
| Peptide, recombinant protein | GST-HIV-1C-PYRE p6 | This paper | | Recombinant protein purified using Glutathione agarose beads from lysates of *E. coli* expressing p GEX-6Pi-C + PYRE p6 described in this Table |
| Peptide, recombinant protein | CCL2 | PeproTech | 300-04-50UG | 250 ng/mL |
| Peptide, recombinant protein | rhM-CSF | PeproTech | AF-300–25 | 50 ng/mL |
| Peptide, recombinant protein | PHA | Sigma | 11249738001 | 1 µg/ml |
| Peptide, recombinant protein | IL-2 | PeproTech | 200–02 | 50 U/ml |
| Commercial assay or kit | HCYTO-MAG kit | Millipore | HCYTOMAG-MCP-1 | For CCL2 measurement via Luminex |
| Commercial assay or kit | Quick Change kit | Agilent Technologies | 210518–5 | For site directed mutagenesis |
| Commercial assay or kit | Luciferase assay kit | Promega | PRE1531 | For measuring the luciferase activity in TZMbl cell lysates |
| Commercial assay or kit | HIV-1 p24 antigen capture assay ELISA | ABL, Inc | 5447 | For p24 measurement |

*Continued on next page*

*Continued*

| Reagent type (species) or resource | Designation | Source or reference | Identifiers | Additional information |
|---|---|---|---|---|
| Software, algorithm | QuantStudio Real-Time PCR software v1.3 | Thermofisher Scientific | | For CT value analysis |
| Software, algorithm | Volocity 6.3 | Perkin Elmer | RRID:SCR_002668 | confocal image analysis |
| Software, algorithm | Graphpad Prism | Graphpad Software, Inc | RRID:SCR_002798 | Plotting data on graphs |
| Software, algorithm | FIJI | NIH | RRID:SCR_002285 | Confocal image analysis |
| Software, algorithm | LI-COR image studio software | LI-COR Biosciences | RRID:SCR_015795 | Western blot band intensity measurements |
| Chemical compound, drug | Gentamicin | Invitrogen | 15750–060 | 0.2% in PBMCs/MDM culture media |
| Chemical compound, drug | Ciprofloxacin | Sigma | 17850 | 4 µg/ml in MDM culture media |
| Chemical compound, drug | Penicillin-Streptomycin | Fischer Scientific | 15140–122 | Provided at 100x, used at 1x concentration in cell culture media (equivalent to 50 to 100 IU/mL Penicillin and 50 to 100 µg/mL Streptomycin) |
| Other | Latrunculin A | Life Technologies | L12370 | 100 ng/mL |
| Other | Texas Red Phalloidin | Life Technologies | T7471 | 1:500 |
| Other | Prolong gold - anti-fade DAPI | Fisher scientific | p36931 | For nuclear staining and mounting |
| Strain, strain background (*E. Coli*) | BL21 (DE3) | Novagen | 69450 | For bacterial expression of pGEX plasmids |
| Biological sample (*H. sapiens*) | Leukopak | New York Blood Center | | New York Blood center |

## Cells, cell lines and viruses

All cell lines were obtained from either ATCC or from the NIH AIDS Reagent Repository and were authenticated at source. Absence of Mycoplasma contamination was verified every 6 months.

Human MDMs were derived from one of two sources. For some experiments, monocytes isolated *via* counter-current elutriation technique, were obtained from the University of Nebraska Medical Center (UNMC). For other experiments, they were obtained *via* differentiation from isolated peripheral blood mononuclear cells (PBMCs) obtained from New York Blood Center. Elutriated monocytes were differentiated into MDMs using Dulbecco's Modified Eagle Medium (DMEM) (GE Healthcare) supplemented with 10% heat inactivated human serum (Sigma), 1% L-glutamine (Invitrogen), 0.2% Gentamicin (Invitrogen), 10 µg/mL Ciprofloxacin (Sigma) and rhM-CSF (5 ng/ml, Peprotech) for 7 days. The PBMC-derived monocytes were differentiated into MDMs using the same media with 50 ng/mL rhM-CSF. PBMCs were isolated from leukopaks obtained from New York Blood center using sequential centrifugation method (*Repnik et al., 2003*) and, activated with PHA (1 µg/ml) for 3 days and maintained in RPMI (GE Healthcare) supplemented with IL-2 (50 U/ml), 10% FBS (Gibco), 10% heat inactivated Human Serum (Sigma), 1% Penicillin-Streptomycin (Gibco), 1% L-Glutamine, and 10 mM HEPES-NaOH pH 7.5.

Primary astrocytes were provided by Dr. Eliseo Eugenin (University of Texas Medical Branch, Galveston, Texas) and were maintained in DMEM (GE Healthcare) containing 10% FBS (Gibco), non-essential amino acids (Gibco), 1% L-glutamine (Gibco) and 1% Penicillin/Streptomycin (Gibco).

HeLa, 293T and TZM-bl cells were cultured in DMEM (GE health care) supplemented with 10% FBS (Gibco), 1% Penicillin-Streptomycin (Gibco), 1% L-Glutamine and 10 mM Hepes-NaOH pH 7.5. CEM.NK$^R$ CCR5 (CEM-CCR5) cells were obtained from the NIH AIDS repository and maintained in RPMI (GE Healthcare) supplemented with 10% FBS (Gibco), 1% Penicillin-Streptomycin (Gibco), 1% L-Glutamine (Invitrogen) and 10 mM HEPES-NaOH pH 7.5.

U87 CD4+ CCR2+ cells referred to as U87-CCR2 (*Björndal et al., 1997*) are from NIH AIDS reagent repository and they are human astrocytoma cells from the parental cell line U87MG. They have been stably transduced with MV7 neo-T4 retroviral vector (G418 resistance) and pBABE-puro-CCR2 (puromycin resistance). These cells constitutively express CCR2 and are grown and maintained in DMEM (Gibco), 15% FBS (Gibco), supplemented with 1 µg/mL puromycin (Sigma) 300 µg/mL G418 (Sigma), 1% L-Glutamine (Gibco) and, 1% Penicillin/Streptomycin (Gibco).

## Generating infectious HIV-1 particles

To generate viruses, the molecular clone DNAs of HIV-1$_{ADA}$ (HIV-1B) (*Theodore et al., 1996*), HIV-1$_{IndieC1}$ (HIV-1C) (*Mochizuki et al., 1999*) or their late domain variants were transfected into $8 \times 10^6$ HEK293T cells using Lipofectamine 3000 kit (Life Technologies). At 6 hr post-transfection, Opti-MEM media was replaced with DMEM supplemented as before. The supernatants, containing high titer infectious virus, were collected at 24 hr post-transfection, p24 was measured via p24 ELISA (see below) and aliquots were frozen for further use. Infectivity of virus particles thus produced was tested using TZMbl cells. These viruses were used for infection of MDMs, PBMCs, CEM-CCR5 or TZMbl cells. For multiday HIV-1 replication assay in MDMs, PBMCs or CEM-CCR5, equal multiplicities of infection (MOI - 10ng p24 per $1 \times 10^6$ cells) were employed for each virus being tested.

## Measuring virus production

### Monocyte-derived macrophages (MDMs)

About $1.5 \times 10^7$ MDMs were infected with HIV-1B or HIV-1C virus at 10ng p24/$10^6$ cells. In order to modulate the levels of CCL2 in the macrophage media, the infected macrophages were treated with: untreated, CCL2 (250 ng/mL), α-CCL2 (2.5 µg/mL) or an isotype control antibody (2.5 µg/mL). Media from the various treatments were collected and 50% replenishment was done every 3 days throughout the multi-day replication experiment up to 30 days. Virus replication was monitored by measuring p24 via ELISA as described under p24 ELISA.

To ascertain the effects on virus release in macrophages, at day 15, HIV-1 particles were prepared by first clarifying the media as mentioned above followed by pelleting the virus through a 20% sucrose cushion at 100,000 Xg, 4°C for 135 min (*von Schwedler et al., 2003*). Cells were lysed and the lysates used for p24 Immunoblotting.

Levels of CCL2 in the media were quantified as described under section 'measurement of CCL2 concentration'.

### CEM-CCR5 cells

To test the effect of CCL2 addition, CCL2 immuno-depletion or *ALIX* silencing on HIV-1B and HIV-1C replication in T cells, we used CEM-CCR5. We plated $1 \times 10^6$ cells/mL in six-well plates, infected with 10ng p24/$10^6$ cells and subjected the cells to the following treatments: untreated control, CCL2, α-CCL2 and Isotype antibody. We rocked the plates gently, incubated them at 37°C, we collected and replenished the media every 3 days until 24 days post-infection. Collected media were centrifuged at 2000 Xg, 4°C for 10 min, we transferred the supernatants collected into new tubes, aliquoted them and stored at −80°C until p24 ELISA measurement. For measurement of CCL2 in CEM-CCR5 cell media, culture supernatants collected every 3 days up to day 15 were employed. Levels of CCL2 in the media were quantified as described under section 'measurement of CCL2 concentration'.

### CCL2 rescue experiment

We differentiated macrophages from PBMCs as before for 7–10 days. We infected macrophages with 10 ng per $1 \times 10^6$ cells and subjected them to the following treatments: untreated control, CCL2 (250 ng/mL), anti-CCL2 (2.5 µg/mL), Isotype antibody (2.5 µg/mL) and, a second α-CCL2 sample (2.5 µg/mL) (for rescue). The media were collected every 3 days and replenished with corresponding treatments. At day 9, for one set of α-CCL2-treated cells, we changed the media and instead of α-CCL2, we added CCL2 (250 ng/mL) and continued the replication experiment until 24 days. Media were centrifuged as before, stored at −80°C. Levels of p24 in the media collected were measured by p24 ELISA.

## HeLa cells

Approximately $5 \times 10^5$ HeLa cells, seeded in six well plates and grown overnight, were transfected with 2 µg/well HIV-1 DNA (HIV-1B$_{ADA}$ or HIV-1C$_{IndieC1}$ molecular clones or their late domain mutants) using Lipofectamine 3000 kit and OptiMEM media (Life Technologies). At 6 hr post-transfection, OptiMEM was replaced with complete DMEM (10% FBS (Gibco), 1% Pen-Strep (Gibco), 10 mM HEPES (pH 7.5). At the time of media change, the transfected cells were either untreated (controls), treated with CCL2, α-CCL2 or isotype antibody. At 24 hr post-transfection, the virus supernatant was collected and centrifuged at 2000 x g at 4°C for 10 min and stored as aliquots at −80°C. Cell lysates of transfected cells were also collected. Cells were lysed using lysis buffer (100 mM HEPES-NaOH pH 7.5, 142 mM KCl, 1.0% NP40 and protease inhibitor cocktail) and clarified by centrifugation at 20, 800 x g for 15 min. In order to ensure that transfections were equivalent between the various samples within the experiment, a GFP expression plasmid peGFP n1 (Clontech) was co-transfected and the cells examined for GFP expression using epifluorescence microscopy. For quantification of CCL2, HeLa cells that were not transfected with HIV were used, but that were untreated, CCL2- or α-CCL2-treated.

For determining effects on virus release, viruses produced in both MDMs and HeLa cells, sucrose cushion purified viruses and cell lysates were resolved in 12% SDS-PAGE gel and transferred to nitrocellulose for Gag-p24 western immunoblot analysis.

## p24 ELISA

For p24 determination, supernatants of HIV-1-infected or transfected cells were collected and centrifuged as before. The cells were pelleted at 4°C, 2000 xg for 10 min, washed 2x with 1X PBS and lysed with Lysis buffer (10 mM HEPES pH 7.4, 142.5 mM KCl, 0.2% NP40 and protease inhibitor cocktail (Roche)). Using appropriate dilutions (in complete DMEM) of the supernatant, the p24 HIV-1 antigen levels were quantified via HIV-1 p24 antigen capture assay ELISA (ABL) to generate standard curve.

## Measurement of CCL2 concentration

CCL2 concentration in the supernatants was quantified using beads (HCYTO-MAG kit) from Millipore (Billerica, MA), measured using a Luminex xMAP Magpix (Luminex Corp, Austin, TX) and analyzed using StarStation (Applied Cytometry Systems, Sacramento, CA).

## HIV-1 sequence analysis

Gag-p6 amino acid sequences for both clades B and C, were retrieved using complete genome sequences from the HIV sequence database at Los Alamos National Laboratory (*Figure 1—figure supplement 1A*). We were able to gather 2012 and 674 full-length HIV-1 clade B and C, respectively. To minimize redundancy, the most variable Gag-p6 sequences were clustered by sequence identity using the program CD-HIT (*Li and Godzik, 2006*). Considering that intra-clade genetic variation tends to be at around 5% we chose to cluster the Gag-p6 sequences at 95% sequence similarity which led to 695 and 475 sequences for B and C, respectively. Gag-p6 protein sequences were subjected to multiple sequence alignments using Clustal Omega at the EMBL-EBI site (*Goujon et al., 2010*; *Sievers et al., 2011*). Difficult sections within the Gag-p6 protein sequences were manually inspected and corrected to ensure proper conservation of the LYPX motif.

The observed numbers in *Figure 3B* were derived by counting the occurrences of the Y amino acid at position 9, in the context of LYPX motif, within the sequences and their alignments found in *Figure 3—source data 1*, *Figure 3—source data 2* and *Figure 3—source data 3*.

## Silencing *ALIX*

Lentivirus-mediated delivery of shRNA was employed for silencing *ALIX* in CEM-CCR5 cells. Lentiviral supernatants were prepared by co-transfecting 293 T cells with pLKO lentiviral packaging plasmid mix (pMission Gag-Pol and pMission VSV-G) and a transfer vector plasmid (pLKO.1 expressing either a scrambled shRNA, the anti-*ALIX* shRNA 29394 (5′ CCGGGCTGCTAAACATTACCAGTTTCTCGA-GAAACTGGTAATGTTTAGCAGCTTTTT 3′) or 29395 (5′ CCGGCCTGAATTACTGCAACGAAATC TCGAGATTTCGTTGCAGTAATTCAGGTTTTT 3′) (*Eekels et al., 2011*) using calcium phosphate method. About 40 hr post-transfection, media were collected, filtered through 0.22 µm filters,

subjected to DNase one treatment and aliquotted for storage and future use. The p24 levels were determined with p24 ELISA (ABL Inc). Equal viral input was used to transduce CEM-CCR5 cells for 48 hr. Transduced cells were selected for with puromycin (1 µg/mL) and *ALIX* knockdown was confirmed with western blot.

## Constructing viruses with Gag p6 mutations

The HIV-1$_{ADA}$ molecular clone was employed to create the HIV-1B mutants Gag p6$^{\Delta LY}$, Gag p6$^{\Delta PTAP}$ $^{(+LIRL)}$ and Gag p6$^{\Delta PTAP(+LIRL), \ \Delta LY}$ and the HIV-1$_{IndieC1}$ molecular clone was used to create HIV-1C mutants, Gag p6$^{+LY}$, Gag p6$^{\Delta PTAP(+LIRL), \ +LY}$ and Gag p6$^{\Delta PTAP(+LIRL)}$. For creating HIV-1B$_{ADA}$ p6 mutants, an intermediate construct was first constructed containing a 4061 bp Age1-EcoR1 fragment spanning *gag* and some portion of *pol* in the plasmid vector pSL1190. For creating HIV-1C$_{IndieC1}$ mutants, we generated an intermediate construct containing a 3818 bp Sca1-Sca1 fragment spanning *gag* and *pol* using the plasmid pBluescript-KS. Site-directed mutagenesis was performed using the QuikChange kit (Agilent Technologies) using primers shown in *Table 1*. The presence of the intended mutations and the absence of undesirable mutations were confirmed by sequence analysis.

## Infectivity of HIV-1 molecular clones and L-domain mutants

About $2 \times 10^5$ TZMbl cells per well were seeded in 12 well plates and incubated for 24 hr at 37°C. Cells were infected with HIV equivalent to 5ng p24 and incubated. After 24 hr, cells were washed and lysed with 1X passive lysis buffer (Promega). Lysates were centrifuged at 20,817 Xg, 4°C, for 15 min. Supernatants were collected and 50 µL lysate was subjected to luciferase assay using Luciferase assay kit (Promega).

## Real-time polymerase chain reaction

For real-time PCR quantitation of *ALIX* or *ccr2b* mRNAs, RNA extracted from HeLa cells (for *ALIX*) or MDMs, CEM-CCR5 or HeLa cells (for *ccr2b*) using RNeasy mini kit was quantified via nanodrop and 2 µg RNA was used to synthesize cDNA using Moloney Murine Leukemia Virus RT enzyme. Quantitative PCR was run on 10 ng of cDNA. For *ALIX* mRNA quantitation, qPCR was performed using 2X Taqman master mix (Applied Biosystems), with cycling conditions was as follows; 50°C for 2 min, 95°C for 2 min, 40 cycles of denaturation at 95°C for 15 s, annealing at 55°C for 15 s, and extension at 72°C for 1 min. For, ccr2b mRNA, quantitative PCR was run using, 2X PowerUP SYBR Green master mix (5 µL), forward and reverse primers (800 nM each), and nuclease-free water. The cycling condition used was as follows: 50°C for 2 min, 95°C for 2 min, 40 cycles of denaturation at 95°C for 15 s, annealing at 55°C for 15 s, and extension at 72°C for 1 min. The amplification and CT values were analyzed using QuantStudio Real-Time PCR software v1.3.

**Table 1.** Sequences of primers employed in site-directed mutagenesis.

| Primer | Sequence |
| --- | --- |
| ADA (PTAP to LIRL mutation) primer **Forward** | 5'CTACAGAGCAGACCAGAGCTGATCAGACTGCCAGAAGAGAGCTTCAGG3' |
| ADA (PTAP to LIRL mutation) primer **Reverse** | 5'CCTGAAGCTCTCTTCTGGCAGTCTGATCAGCTCTGGTCTGCTCTGTAG3' |
| ADA (LY deletion) primer **Forward** | 5'GGAGCCGATAGACAAGGAACCTTTGACTTCCCTC3' |
| ADA (LY deletion) primer **Reverse** | 5'GAGGGAAGTCAAAGGTTCCTTGTCTATCGGCTCC3' |
| IndieC1 (PTAP to LIRL mutation) primer **Forward** | 5'CTCCAGAGCAGACCGGAGC**TGA**T**CAGA**CTGCCAGCAGAGAGCTTCAGG3' |
| IndieC1 (PTAP to LIRL mutation) primer **Reverse** | 5'CCTGAAGCTCTCTGCTGGCAGTCTGATCAGCTCCGGTCTGCTCTGGAG3' |
| IndieC1 (LY insertion) primer **Forward** | 5'GCCGAAAGACAGGGAACTGTATCCCTTAACTTCCCTCAAA3' |
| IndieC1 (LY insertion) primer **Reverse** | 5'TTTGAGGGAAGTTAAGGGATACAGTTCCCTGTCTTTCGGC3' |

DOI: https://doi.org/10.7554/eLife.35546.045

## Analysis of total, soluble and insoluble ALIX levels in cells

Approximately $4 \times 10^6$ HeLa cells were plated and incubated overnight at 37°C. Cells were treated with CCL2, α-CCL2, nothing (untreated) or Isotype antibody control for 24 hr. 1 hr before collecting the cells, we added either DMSO only (control) or Latrunculin A to cells. After 1 hr, cells were collected and lysed (100 mM HEPES-NaOH, pH 7.5, 142.5 mM KCl, 1% Triton X-100, 5 mM MgCl2, 1:100 dilution of phosphatase inhibitor cocktail I (microcystine LR, cantharidin and (-)-p-bromotetramisole), phosphatase inhibitor II (sodium vanadate, sodium molybdate, sodium tartrate and imidazole) Sigma and, Protease inhibitor cocktail). The lysates were divided into two fractions: one used directly for analysis of total lysates and the other centrifuged at 20,817 Xg, 4°C for 15 min. Soluble fractions were collected while pellets (termed pellet-1) were not studied further as they consist of nuclei, cellular debris and cross-linked actin. The soluble fraction was subjected to ultracentrifugation at 95,000 Xg at 4°C. The supernatant after ultra-centrifugation was collected as the soluble fraction to be analyzed, the ultra-centrifuged pellet was washed 2X with cold PBS, resuspended in 5X sample buffer as pellet-2 for analysis. The total lysate, soluble and pellet-2 were subjected to Western immunoblot analysis. The blots were probed with α-ALIX mouse (Thermo-Scientific), α-TSG101 mouse (GeneTex), anti-β-actin mouse (Thermo-Scientific), in addition to α-GAPDH mouse (Thermo-Scientific) as loading control.

## Western immunoblotting

For western immunoblotting, 5X sample buffer was added to bring all samples to 1X and boiled for 5 min. All samples were loaded and resolved on 12% SDS-PAGE gels. The gels were transferred to nitrocellulose membrane and blocked with 5% non-fat milk (Bio-Rad) in 1X TBS-T buffer or 5% BSA (Sigma) in 1X PBS-T buffer (Blocking buffer) for 1 hr at room temperature. Membranes were incubated with primary antibodies overnight at 4°C in Blocking buffer, washed with 1X TBS-T or PBS-T buffer and then incubated with secondary antibodies for 1 hr at room temperature. After the final wash with 1X TBS-T or PBS-T, membranes were developed using chemiluminescent peroxidase substrate (Pierce ECL from Thermo Scientific) or Odyssey Fc *LI-COR*. Quantitation of the bands was done via FIJI or *LI-COR* image studio software.

For Gag-p24 immunoblot analysis, membranes were probed with anti-HIV p24 goat (1:5000) (Kind gift of Dr. David Ott, NCI, Frederick) and anti-GAPDH-mouse (Thermo Scientific; 1:10,000) antibodies overnight at 4°C. We employed Donkey α-Goat antibodies labeled with Alexa 800 nm (Life Technologies; 1:10,000) and Goat α-mouse antibodies labeled with Alexa 680 nm (Life Technologies; 1:10,000) as secondary antibodies. Protein bands were quantified Using *LI-COR* (Odyssey-fc).

For immunoblotting CCR2, we collected and lysed U87-CCR2, primary astrocytes, CEM-CCR5, HeLa cells and macrophages. After quantifying protein concentration with BSA assay (Thermo Scientific), we mixed the lysates with 5X sample buffer, boiled the samples, spin briefly and resolved the samples on 12% SDS PAGE followed by Immunoblotting as described before.

## GST pull-downs

The GST protein and GST-p6 fusion proteins from HIV1B and HIV-1C were expressed in *E. coli* and purified from 1 L cultures of BL21DE3-pGEX-6Pi, pGEX-6Pi-HIV-1Bp6, pGEX-6Pi-HIV-1Cp6, pGEX-6Pi-HIV-1Cp6-PYKE and pGEX-6Pi-HIV-1Cp6-PYRE. They were grown to an $A_{600}$ of ~0.8 before induction with 1 mM IPTG for 5 hr at 37°C. The bacterial pellet from 1 L of culture was resuspended in 10 mL of Buffer Y (low salt) (50 mM Tris pH8.0, 50 mM NaCl, 1 mM EDTA, 0.5% IGEPAL, Protease inhibitor tablet (Roche)). The bacterial suspension was subjected to seven freeze-thaw cycles using dry ice/ethanol. Lysozyme was added to a final concentration of 10 mg/mL and incubated for 30 min at 4°C before adding sufficient 5M NaCl to increase final NaCl concentration to 500 mM. Lysates were incubated 1 hr at 4°C, then centrifuged at 27,000 xg, 4°C for 30 min. The supernatant was bound to G-beads (GE health care) and quantitated on a 12% SDS PAGE gel, with known concentration of BSA as a quantitation reference standard. The gel was stained with Coomassie brilliant blue and destained. Purified GST, GST-HIV-1Bp6, GST-HIV-1Cp6 and, GST-HIV-1C-PYxE p6 proteins bound to G-beads were stored at 4°C.

Soluble cell lysate fractions from HeLa cells, prepared as described above, treated with CCL2 (250 ng/mL) or α-CCL2 (1 µg/mL) were prepared and quantitated using Micro BCA protein assay kit (Thermo Scientific). The expression of ALIX and TSG101 were first confirmed by Western blot. Equal

inputs of GST-tagged proteins bound to G-beads were incubated for 1 hr at 4°C with normalized amounts HeLa cells lysates expressing ALIX and TSG101 in binding buffer (200 mM HEPES-KOH pH 8.0, 50 mM DTT, 1% IGEPAL, 100 mg/mL BSA and protease inhibitors). Prior to incubation, 40 μL of the binding reaction was collected and used for input determination. Following incubation, the beads were washed 5–7 times with 1X wash buffer (50 mM Tris-Cl pH 8.0, 0.1 mM EDTA, 200 mM NaCl, 1.0% NP-40, 25 mM PMSF). Bound proteins and input were separated by 12% SDS-PAGE, subjected to either immunoblotting or Coomassie brilliant blue. Western blots were analyzed using anti-ALIX mouse (1:500) (Thermo Scientific) and α-TSG101 mouse (1:500) (GeneTex) antibodies. The relative binding was quantified by densitometry (ImageJ).

## Fluorescence microscopy

HeLa and MDM cells were seeded on MatTek plates and incubated at 37°C, 5% $CO_2$ overnight. Cells were treated with CCL2 (250 ng/mL), α-CCL2 (2.5 μg/mL), 1X PBS or isotype control antibodies (2.5 μg/ml) for 4 hr. Cells were washed twice with 1X PBS and fixed with 4% Paraformaldehyde (Alfa Aesar) for 15 min at room temperature. Fixed cells were permeabilized with 0.1% Triton-X solution for 3 min and blocked with 5% BSA at 37°C for 1 hr. Cells were incubated with anti-ALIX antibody (Thermo Scientific) at room temperature for 2 hr. Cells were washed and incubated with goat anti mouse-Alexa 488 antibody (1:250; Life Technologies) for 1 hr in the dark. Cells were washed as before and incubated with Texas Red Phalloidin (Life Technologies) for 30 min at room temperature. Prolong-Gold (Invitrogen) mounting media with DAPI was used to overlay glass coverslips. Cells were imaged using Leica SP8 confocal microscope. Image analysis was performed using Volocity 6.3 image analysis software. Co-localization measurements for ALIX and F-actin were performed on 50 cells (HeLa) and 100 cells (MDMs) from 12 different samples for each of the treatments and the average Pearson's Coefficient for each sample was plotted using Prism (*Manders et al., 1992*). The Pearson's coefficients were calculated by analyzing each cell separately in confocal sections.

## Measurement of Gag distribution

For Gag localization confocal imaging, HeLa cells were seeded on cover glasses and, incubated at 37°C overnight. Cells were transfected as before with HIV-1B$_{ADA}$ molecular clones and subjected to one of the following treatments: untreated control, α-CCL2 (2.5 μg/mL) and, Isotype antibody (2.5 μg/mL) until 24 hr post transfection. Media were aspirated, cells washed as before and fixed with Eddy fix buffer (3.7% PFA, 0.1% glutaraldehyde, 0.15 mg/mL saponin in 1X PBS) for 15 min at room temperature. Cells were washed three times with 1X PBS, permeabilized with 1% Triton X-100 for 3 min at room temperature, followed by RNase treatment at 37°C for 45 min. Fixed cells were washed and blocked with 5% BSA blocking buffer for 1 hr at room temperature, cells were washed three times as before, and incubated with primary antibody (anti-p24 goat (1:500)) in blocking buffer for 2 hr at room temperature. Cells were washed three times and incubated with secondary antibody (Donkey anti-goat Alexa 488 (1:500) Life technologies) for 1 hr in the dark. Cells were washed and mounted with anti-fade plus DAPI. Cells were imaged with confocal microscope Leica SP8.

We quantified Gag intensity in 13 cells for each treatment: Untreated, α-CCL2 and Isotype antibody using Image J. Briefly, after setting our line scale to pixel/unit, we drew a line of constant thickness and length. We obtained pixel intensities across the drawn line (plasma membrane towards the nucleus).

## Electron microscopy

About $6 \times 10^5$ HeLa cells were plated per well of 6 well plates and incubated at 37°C overnight. Cells were transfected with HIV-1B$_{ADA}$ molecular clone as before and, incubated at 37°C until media change to complete DMEM. Cells were treated with the following: untreated control, α-CCL2 (2.5 μg/mL) and, Isotype antibody (2.5 μg/mL). Cells were incubated at 37°C until 24 hr post transfection. Cells were washed and fixed with 2.5% glutaraldehyde in 0.1 M sodium cacodylate buffer, post-fixed with 1% osmium tetroxide followed by 2% uranyl acetate for 30 min each, dehydrated through a graded series of ethanol with 5–10 min changes, and then stripped from the 6-well plates using propylene oxide. Cells were then transferred to microcentrifuge tubes and embedded in LX112 resin (LADD Research Industries, Burlington VT). Ultrathin sections were cut on a Reichert Ultracut UC7,

stained with uranyl acetate followed by lead citrate and viewed on a JEOL 1200EX transmission electron microscope at 80kv. Images were collected and analyzed.

### Replicates and statistics

We define biological replicates as experiments that were performed on two or more separate days (often separated by weeks to months) employing cells derived from different human donors (as in the case of MDMs or PBMCs) or even a cell line employed at different passages on different days as above. In our data, we have either combined multiple biological replicates (as indicated) to plot the data as a single set along with SEM or depicted representative data from one of the several biological replicates (indicated for each experiment). Technical replicates in our experiments are replicates within one experiment with no biological differences in the replicates. Statistical significance was determined by performing Student's $t$ test and area under the curve.

## Acknowledgements

The authors wish to acknowledge Duncan Wilson for many useful discussions, Kartik Chandran and Kimberly Merani for reading the manuscript, Eitan Novogrodsky for technical assistance, the Einstein-Rockefeller-CUNY Center for AIDS Research (ERC-CFAR) (P30 AI051519) for the use of core facilities and Dr. Eliseo Eugenin for the use of confocal microscope at Rutgers University in our initial studies. The authors would like to acknowledge the Analytical Imaging Facility (AIF) for assistance with microscopy: Hillary Guzik for assistance with the confocal microscope (Leica SP8 was funded by Shared Instrumentation Grant 1S10OD023591-01) and figures and Timothy Mendez for assistance with the electron microscopy (JEOL JEM-1400Plus was funded by a Shared Instrumentation Grant 1S10OD016214-01A1). The AIF is funded in part by the National Cancer Institute Cancer Grant P30CA013330. The work reported here was supported by NIH grants R37 AI030861 and R01 MH083579 (to VRP) and R01 GM112520 (to GVK). DOA acknowledges support from the institutional HIV/AIDS training grant NIH T32 AI007501 and F31 AI127295. The authors declare that they have no conflicts of interest.

## Additional information

### Funding

| Funder | Grant reference number | Author |
| --- | --- | --- |
| National Institutes of Health | R37 AI030861 | Vinayaka R Prasad |
| National Institutes of Health | R01 MH083579 | Vinayaka R Prasad |
| National Institutes of Health | R01 GM112520 | Ganjam V Kalpana |
| National Institutes of Health | T32 AI007501 | David O Ajasin |
| National Institutes of Health | F31 AI127295 | David O Ajasin |
| National Institutes of Health | T32 GM007491 | Arthur P Ruiz |

The funders had no role in study design, data collection and interpretation, or the decision to submit the work for publication.

### Author contributions

David O Ajasin, Formal analysis, Investigation, Visualization, Methodology, Writing—original draft, Writing—review and editing; Vasudev R Rao, Formal analysis, Investigation, Methodology, Writing – review and editing; Xuhong Wu, Investigation; Santhamani Ramasamy, Arthur P Ruiz, Investigation, Writing—review and editing; Mario Pujato, Formal analysis, Investigation; Andras Fiser, Formal analysis, Supervision, Investigation; Anne R Bresnick, Supervision, Methodology, Writing—review and editing; Ganjam V Kalpana, Supervision, Investigation, Methodology, Writing—review and editing; Vinayaka R Prasad, Conceptualization, Formal analysis, Supervision, Funding acquisition, Investigation, Methodology, Writing—original draft, Project administration, Writing—review and editing

## Author ORCIDs

David O Ajasin (iD) https://orcid.org/0000-0003-1061-1860
Vasudev R Rao (iD) https://orcid.org/0000-0002-9435-4023
Vinayaka R Prasad (iD) https://orcid.org/0000-0002-9461-0189

## Decision letter and Author response

Decision letter https://doi.org/10.7554/eLife.35546.046
Author response https://doi.org/10.7554/eLife.35546.047

## Additional files

### Data availability

All data generated or analysed during this study are included in the manuscript and supporting files. Source data files have been provided for Figures 1, 2, 3A, 3B, 4B, 4C, 4E, 5A, 5B, 6B, 7B, 8B, 10A, 10B, 11B, 12B, 12C and 13, Figure 1—figure supplement 1A, 1B, 1C and 1D, Figure 1—figure supplement 2A, Figure 1—figure supplement 3 and Figure 4—figure supplement 2.

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
