## [Decision Letter]

[Editors’ note: this article was originally rejected after discussions between the reviewers, but the authors were invited to resubmit after an appeal against the decision.]

Thank you for submitting your work entitled "CCL2 Mobilizes ALIX to Facilitate Gag-p6 Mediated Virion Release" for consideration by *eLife*. Your article has been reviewed by three peer reviewers, and the evaluation has been overseen by a Reviewing Editor and a Senior Editor. The reviewers have opted to remain anonymous. As you will see below, we have concluded that your paper will not be considered further at *eLife*.

Ajasin et al. propose that binding of the host factor ALIX to its "YPXL" binding site on HIV-1 Gag-p6 is functionally modulated by the chemokine CCL2, and that this regulatory axis promotes replication of clade B HIV-1 isolates. Specifically, the authors argue that CCL2 promotes releases ALIX from the actin cytoskeleton, thereby increasing the ALIX-p6 interaction and stimulating HIV-1 budding. Unlike other HIV-1 clades, most subtype C isolates lack a functional ALIX binding site in p6. Nicely, the authors found that a subtype C virus was insensitive to CCL2 but became responsive when an ALIX binding site was introduced, which strengthens the case that CCL2 is acting via ALIX.

The activity and mechanism by which CCL2 enhances HIV-1 replication is of interest, which makes the study potentially important. More generally, the effects of extracellular stimuli on virus exit (except for those of IFNs) have not been widely studied. Despite intriguing observations, however, a number of aspects of the study are not yet convincing because several of the key observations need to be established more rigorously and because the mechanism underlying CCL2 alteration of ALIX subcellular localization is rather preliminary and requires more complete characterization. Given the quantity of experimental work required for publication, the recommended decision is rejection. However, the authors would be welcome to resubmit a new, improved manuscript to *eLife* on this topic if they are able to address the reviewers' concerns (described below). Alternatively, the authors are, of course, welcome to submit the manuscript to another journal if they prefer.

Issues that must be addressed:

1) The significance and reproducibility of the majority of the virological data must be better established. Most of these experiments are represented as mean +/- SEM (n = 2). Moreover, biological replicates of some of the experiments have apparently been performed only twice (or even once). This issue may underlie the inconsistency of some of the described phenotypes. For example, addition of anti-CCL2 antibodies has a 40-fold effect in Figure 1B, whereas the same treatment has a 2-fold effect in Figure 1E. All of the experiments must be done independently at least three times. Finally, HIV-1 replication in MDM can vary significantly from donor to donor. The authors should state whether the experiments shown were repeated using different donors (and if they not, then this must be done).

2) Figure 1: The effect of adding extra CCL2 on subtype B HIV-1 is considerably less impressive than the effect of anti-CCL2. Can virus replication in the presence of anti-CCL2 be rescued by addition of exogenous CCL2? This would confirm that the effect of the antibody is due to the neutralization of endogenous CCL2, rather than an off-target effect. In a similar vein, these experiments need to be better controlled. An isotype antibody control should be used in these experiments and specificity should also be shown at the chemokine level (e.g., are similar phenotypes observed when other chemokines are inhibited or added to these cells?).

3) The author's model indicates that addition of CCL2 should disassociate ALIX from actin but this effect is not observed in Figure 4, thus contradicting some of the main conclusions in the manuscript. These observations are further contradicted by the data in Figure 3B, where addition of anti-CCL2 antibodies does not alter the fraction of soluble ALIX. Moreover, the authors should examine whether the insoluble pellets in Figure 3B actually represent F-actin. For example, does actin in the pellet disappear upon latrunculin treatment? If so, does pelletable ALIX also diminish upon latrunculin treatment under all three conditions (UT, CCL2, and anti-CCL2)? Finally, the observed changes in ALIX localization upon CCL2 depletion are intriguing but require greater mechanistic explanation as meaningful associations between ALIX and the actin cytoskeleton remain to be defined more rigorously, and the mechanism underlying CCL2-dependent release of ALIX from actin also requires greater definition.

4) The authors have observed CCL2-responsiveness in macrophages, the T cell line SupT1, and in primary CD4 T cells. For example, Figure S3 shows that the replication of a clade B but not a clade C virus in SupT1 cells was inhibited by anti-CCL2. However, RNAseq data indicate that SupT1 cells express minimal amounts of CCL2 (as do other T cell lines). Similarly, RNAseq data indicate that purified primary CD4 T cells often express little or no CCL2, even after infection with HIV-1. The authors should indicate whether and how much CCL2 was expressed in the experiment involving SupT1 cells and primary CD4 T cells. Furthermore, the purity of the primary CD4 T cells should be demonstrated, to rule out the possibility that CCL2 came from contaminating monocytes/macrophages.

5) Figure S3: the p24 values shown are extremely low (pg rather than ng), indicating negligible virus replication in SupT1 cells, which in turn raises concerns about the relevance of the data. Although SupT1 cells are extremely permissive for X4 viruses, they express hardly any CCR5, and most R5 viruses cannot infect them. Indeed, Dejuco et al. have reported that ADA, one of the viruses used in the present study, does not infect SupT1 cells (J Virol 73, 7842, 1999). Although the authors may have used SupT1/CCR5 cells, this is not indicated anywhere. The Materials and methods merely state that the cells were from the NIH AIDS repository, which are unmodified.

6) Figure 1C and 6C: The effect of anti-CCL2 (i.e., accumulation of Pr55 and reduction of mature p24) seems more typical of an assembly defect (e.g., membrane binding or multimerization) than of the particle release defect. Have the authors confirmed that cells treated with anti-CCL2 accumulate arrested budding virus particles on the cell surface?

7) Figure 2B, C, E, 7C, and 8: These panels, which show the levels of released p24, should be accompanied by immunoblotting of cell lysates. This would confirm that Gag expression levels in cells are comparable under the different conditions tested.

8) The discussion should be strengthened in several respects. A) Considering that the main novelty/significance of this study comes from the finding that CCL2 triggers ALIX dissociation from actin cytoskeleton, additional mechanistic explanation (even if speculative) is desirable. In this regard, the discussion on RhoA is probably irrelevant, since RhoA regulates actin cytoskeleton assembly, whereas CCL2 does not seem to affect actin cytoskeleton formation (based on the amount of actin associated with insoluble pellets (Figure 3) or the abundance of stress fibers (Figure 4)), but rather the release of ALIX from actin. B) The authors state that the finding of CCL2-triggered ALIX redistribution has wide-ranging implications in cell and cancer biology but specific discussion on such implications is not provided.

---

## [Author Response]

[Editors’ note: the author responses to the first round of peer review follow.]

Ajasin et al. propose that binding of the host factor ALIX to its "YPXL" binding site on HIV-1 Gag-p6 is functionally modulated by the chemokine CCL2, and that this regulatory axis promotes replication of clade B HIV-1 isolates. Specifically, the authors argue that CCL2 promotes releases ALIX from the actin cytoskeleton, thereby increasing the ALIX-p6 interaction and stimulating HIV-1 budding. Unlike other HIV-1 clades, most subtype C isolates lack a functional ALIX binding site in p6. Nicely, the authors found that a subtype C virus was insensitive to CCL2 but became responsive when an ALIX binding site was introduced, which strengthens the case that CCL2 is acting via ALIX.The activity and mechanism by which CCL2 enhances HIV-1 replication is of interest, which makes the study potentially important. More generally, the effects of extracellular stimuli on virus exit (except for those of IFNs) have not been widely studied. Despite intriguing observations, however, a number of aspects of the study are not yet convincing because several of the key observations need to be established more rigorously and because the mechanism underlying CCL2 alteration of ALIX subcellular localization is rather preliminary and requires more complete characterization. Given the quantity of experimental work required for publication, the recommended decision is rejection. However, the authors would be welcome to resubmit a new, improved manuscript to eLife on this topic if they are able to address the reviewers' concerns (described below). Alternatively, the authors are, of course, welcome to submit the manuscript to another journal if they prefer.Issues that must be addressed:1) The significance and reproducibility of the majority of the virological data must be better established. Most of these experiments are represented as mean +/- SEM (n = 2). Moreover, biological replicates of some of the experiments have apparently been performed only twice (or even once). This issue may underlie the inconsistency of some of the described phenotypes. For example, addition of anti-CCL2 antibodies has a 40-fold effect in Figure 1B, whereas the same treatment has a 2-fold effect in Figure 1E. All of the experiments must be done independently at least three times. Finally, HIV-1 replication in MDM can vary significantly from donor to donor. The authors should state whether the experiments shown were repeated using different donors (and if they not, then this must be done).

Reproducibility: We do not believe that the reproducibility of the work was an issue. But we understand the reviewers’ concern, which was partly based on the number of replications of some of the experiments. Therefore, to comply with the reviewers’ request, we have mounted a lengthy, major effort to fully address this comment. All experiments in the revised manuscript have been performed at least 3 times. In addition, wherever monocyte-derived macrophages (MDMs) are used, those experiments have now been performed in cells derived from 2-3 different donors. We have indicated the number of times each experiment was replicated in the Figure Legends.

Variable fold-differences: The reviewer noted the discrepancy in fold differences between previous Figures 2B (40-fold) to 2E (2-fold) (not Figure 1E to Figure 1B as stated in the review). The corresponding figures now are Figure 3B and 3E, with the data in 3E having been generated using a different CD4 T cell line, CEM-CCR5 in response to a separate comment (see below). The reasons for the lower (2-fold) reduction in virus production by CEM-CCR5 cells (compare untreated to anti-CCL2 in Figure 3E) is that CEM-CCR5 cells produce miniscule levels of CCL2 as compared to HeLa cells (see response under point #4 about CCL2 levels). It is true that the HeLa cell virus production data in the previous Figure 2B and 2C (now Figure 3B and 3C) showed ~40-fold difference between untreated and anti-CCL2-treated samples. We have now repeated this experiment three more times and present an average of four experiments – the average fold difference in HeLa cells is now 5-fold.

2) Figure 1: The effect of adding extra CCL2 on subtype B HIV-1 is considerably less impressive than the effect of anti-CCL2. Can virus replication in the presence of anti-CCL2 be rescued by addition of exogenous CCL2? This would confirm that the effect of the antibody is due to the neutralization of endogenous CCL2, rather than an off-target effect. In a similar vein, these experiments need to be better controlled. An isotype antibody control should be used in these experiments and specificity should also be shown at the chemokine level (e.g., are similar phenotypes observed when other chemokines are inhibited or added to these cells?).

Differential fold reduction and stimulation by addition of CCL2 versus anti-CCL2 antibodies: We agree that in Figure 1, addition of extra CCL2 has a smaller effect compared to CCL2-depletion. Depletion of CCL2 inhibited virus production ~10-fold in macrophages whereas addition of CCL2 increased it by approximately 50% of untreated control. This asymmetry also applies to the HeLa cell experiments (now Figure 3B) where we observe about 5-fold inhibition of virus production with anti-CCL2, but ~50% increase with CCL2 addition. On the other hand, in T cell lines – as seen in the new Figure 2B or 3E, CCL2 depletion results in a 2-fold inhibition, while the addition of CCL2 still causes only an ~50% increase. It is possible that there is an upper limit on how much CCL2 can increase virus production as is observed in our experiments. This could be due to the limitations in the components involved. For example, the level of ALIX could be constant and limiting. Thus, the added CCL2 will be able to mobilize the population of ALIX molecules present in the cytoskeleton causing a proportional increase in virus release. On the other hand, when CCL2 is depleted, most of the soluble ALIX is directed to cytoskeleton, preventing the availability of ALIX to a greater extent, thus leading to a greater level of inhibition. In addition, there may be a variability in: (i) the levels of endogenous CCL2 and (ii) total levels of ALIX present in different cell types. While the addition of anti-CCL2 is able to redirect most of the soluble ALIX to the cytoskeletal compartment and reduce virus production levels, addition of CCL2 to a cell that already has significant endogenous levels of CCL2 can only redirect cytoskeletally associated ALIX molecules to the cytoplasm. We believe that our results are consistent with our model and we have added this to the Discussion section.

Rescue by replacing anti-CCL2 with CCL2: As suggested by the reviewers, we have performed an experiment where anti-CCL2 was added to the medium until 9 days post-infection and was replaced with CCL2 for subsequent days to evaluate rescue of virus production in macrophages. Our data indicates that virus release suppression observed during 9 days of replication in MDMs in the presence of anti-CCL2, was rescued continuously up to 24 days by addition of CCL2. This result confirms that the effect of the antibody is due to neutralization of endogenous CCL2 rather than off-target effects of the antibody, and that the effect is reversible.

Isotype control antibodies: An isotype antibody control was included in Figure 1 in the original manuscript. In the revised manuscript, we have included isotype controls in all experiments where needed (Figures 1 to 3, Figure 4B, Figures 5 to 8, Figure 1—figure supplement 2, Figure 4—figure supplement 1 and Figure 12—figure supplement 1).

Chemokine specificity: Our rationale for investigating the role of CCL2 in HIV-1 replication is supported by the literature. Multiple studies have reported the association of CCL2 with viremia in HIV+ individuals. Importantly, Ansari et al. queried over 100 cellular proinflammatory genes and found CCL2 to be the most significantly correlated with plasma viremia (Ansari et al., 2006). Based on the reviewers’ comment, we considered testing multiple chemokines in our assays. But there is a dizzying array of chemokines known. Even limiting to β-chemokines would require testing over two dozen β-chemokines in three different cell types (MDMs, CD4 T cells and HeLa cells). Therefore, while this is a laudable future goal, testing each of the several β-chemokines is beyond the scope of the current manuscript.

3) The author's model indicates that addition of CCL2 should disassociate ALIX from actin but this effect is not observed in Figure 4, thus contradicting some of the main conclusions in the manuscript. These observations are further contradicted by the data in Figure 3B, where addition of anti-CCL2 antibodies does not alter the fraction of soluble ALIX. Moreover, the authors should examine whether the insoluble pellets in Figure 3B actually represent F-actin. For example, does actin in the pellet disappear upon latrunculin treatment? If so, does pelletable ALIX also diminish upon latrunculin treatment under all three conditions (UT, CCL2, and anti-CCL2)? Finally, the observed changes in ALIX localization upon CCL2 depletion are intriguing but require greater mechanistic explanation as meaningful associations between ALIX and the actin cytoskeleton remain to be defined more rigorously, and the mechanism underlying CCL2-dependent release of ALIX from actin also requires greater definition.

Addition of CCL2 must disassociate ALIX from Actin: This is indeed a key element of our model. However, as mentioned above, the change in ALIX distribution from untreated to CCL2-treated condition tends to have an incremental effect on the levels of ALIX in the soluble fraction – which depends on how much ALIX is pre-associated with the actin cytoskeletal compartment in the presence of endogenous CCL2. We do not understand why the reviewer believes that the previous Figure 4 contradicted our model. There is a clear, observable difference in ALIX colocalization in cells treated with anti-CCL2 vs. CCL2 (previously Figure 4, now Figures 5 and 6 for HeLa cells and macrophages respectively). To illustrate this point, we are providing a higher magnification image of a region of interest in the HeLa cell images (Figure 5). One can observe the following in these images. With a-CCL2, ALIX colocalizes with F-actin structures (stress fibers in HeLa cells and podosomes in macrophages), whereas in CCL2-treated HeLa cells or MDMs, ALIX exhibits a more diffuse cytosolic distribution and very little colocalization with F-actin. These observations were also quantitated by calculating the Pearson colocalization coefficient, which provides a quantitative measure of the degree of colocalization between two fluorophores. This data support the conclusion that ALIX localizes to F-actin structures when CCL2 is depleted. Since HeLa cells (and MDMs) secrete CCL2 into the medium, which has autocrine effects on HeLa cells, the differences in ALIX distribution observed in untreated and CCL2-treated cells appears to be minimal. These results are in agreement with the smaller effect of CCL2 addition on virus production (Figure 1 and Figure 2).

Distribution of ALIX in soluble and insoluble fractions: The reviewer indicates that Figure 3B does not show a reduction in soluble ALIX in anti-CCL2 treated cells. While Western blots are not always quantitative, in the previous Figure 3B, there was indeed an observable reduction in the amount of soluble ALIX between the untreated and anti-CCL2 lanes. This effect was reproducible as shown in the newer replicate of this experiment included in the revised manuscript (now Figure 4B).

Do the insoluble pellets actually represent F-actin and does ALIX in the cytoskeletal compartment diminish upon actin depolymerization?: For examining the actin cytoskeleton and associated proteins, the preparation of detergent insoluble fractions is a well-established approach. This fraction is composed of crosslinked actin filaments, myosin-II and associated actin binding proteins (see Rosenberg et al., 1981; Tarone et al., 1984 and Hartwig and Shevlin, 1986). While detergent-resistant cytoskeletons contain crosslinked actin filaments, short actin filaments will remain in the supernatant. To address this, we modified our protocol to evaluate both detergent-resistant F-actin cytoskeletons and soluble F-actin. Cell lysates were first centrifuged at low speed to separate the detergent-resistant F-actin cytoskeletons and the soluble fraction, which contains F-actin that is not crosslinked into the cytoskeleton. The low speed supernatant was then centrifuged at 100,000 x g, which will pellet soluble actin filaments (labeled Pellet 2 in Figure 4B); G-actin will remain in the supernatant. As expected, our results show that Pellet 2 contains actin. In addition, we used Latrunculin A to reduce the amount of F-actin. Consistent with previous observations (Peterson and Mitchison, 2002), Latrunculin A treatment did not lead to a complete dissolution of all F-actin structures (Figure 4B). However, Latrunculin A reduced both the amount of Factin and ALIX in the high speed pellet, indicating that actin and the associated ALIX in the cytoskeleton appear in the soluble fraction. These data provide strong support for CCL2-mediated regulation of ALIX’s association with the actin cytoskeleton.

4) The authors have observed CCL2-responsiveness in macrophages, the T cell line SupT1, and in primary CD4 T cells. For example, Figure S3 shows that the replication of a clade B but not a clade C virus in SupT1 cells was inhibited by anti-CCL2. However, RNAseq data indicate that SupT1 cells express minimal amounts of CCL2 (as do other T cell lines). Similarly, RNAseq data indicate that purified primary CD4 T cells often express little or no CCL2, even after infection with HIV-1. The authors should indicate whether and how much CCL2 was expressed in the experiment involving SupT1 cells and primary CD4 T cells. Furthermore, the purity of the primary CD4 T cells should be demonstrated, to rule out the possibility that CCL2 came from contaminating monocytes/macrophages.

CCL2-responsiveness in CD4 T cells: In response to the reviewers’ comment about the use of different CD4 T cell lines in different experiments, we have now used a single CD4 T cell line – CEM-CCR5 wherever a CD4 T cell line is used. We performed CCL2 measurements in CEM-CCR5 as requested. In agreement with the RNAseq data cited by the reviewer for SupT1 cells, CD4 T cells (CEM-CCR5) secrete very little CCL2 with or without HIV-1B-infection (Figure 1—figure supplement 2). As we have not used primary CD4 T cells in these experiments, we have not measured CCL2 in primary CD4 T cells.

5) Figure S3: the p24 values shown are extremely low (pg rather than ng), indicating negligible virus replication in SupT1 cells, which in turn raises concerns about the relevance of the data. Although SupT1 cells are extremely permissive for X4 viruses, they express hardly any CCR5, and most R5 viruses cannot infect them. Indeed, Dejuco et al. have reported that ADA, one of the viruses used in the present study, does not infect SupT1 cells (J Virol 73, 7842, 1999). Although the authors may have used SupT1/CCR5 cells, this is not indicated anywhere. The Materials and methods merely state that the cells were from the NIH AIDS repository, which are unmodified.

We performed the experiment in Figure S3 using SupT1-CCR5 cells obtained from Dr. James Hoxie (University of Pennsylvania). The reviewer is correct that the virus production was suboptimal, which was due to the low MOI of virus used in the infection. Therefore, we have now reproduced the experiment depicted in Figure S3 using CEM-CCR5 cells (also referred to as CEM NKR-CCR5) and matched the conditions including virus MOI and anti-CCL2 antibody concentration we employed for Macrophage infection – as in the Figure 1A. The new data (Figure 2A and 2B) show a more robust virus production (p24 over 30ng/ml to 50ng/ml range at peak for HIV-1C and HIV-1B respectively) albeit with a reproducibly modest inhibition with anti-CCL2 for the reasons discussed above.

6) Figure 1C and 6C: The effect of anti-CCL2 (i.e., accumulation of Pr55 and reduction of mature p24) seems more typical of an assembly defect (e.g., membrane binding or multimerization) than of the particle release defect. Have the authors confirmed that cells treated with anti-CCL2 accumulate arrested budding virus particles on the cell surface?

We thank the reviewer for this insightful comment. Further experimentation to investigate this question has lead us to conclude that the basis for the absence of virus particle release is a combination of defects in trafficking and plasma membrane binding leading to virion particles in cytoplasmic vacuoles rather than a traditional budding defect where the particles remain attached to the plasma membrane.

In order to address whether particles accumulate near the membrane in the presence of anti-CCL2, we have performed both confocal microscopy and electron microscopy using HeLa cells transfected with HIV- 1B molecular clones. Confocal microscopy combined with p24 signal pixel quantitation from nuclear membrane to plasma membrane of untreated cells typically shows a pattern of higher concentration of Gag at the cell periphery, close to the plasma membrane (Cano and Kalpana, 2011; La Porte and Kalpana, 2016). However, under anti-CCL2 conditions there is a more even distribution of signal density across the cytoplasm suggesting that normal Gag trafficking is disrupted. We have examined 24h samples where we can see virus particles at the cell surface for untreated cells and cells treated with isotype control antibodies. Anti-CCL2-treated samples at 24 h, however, showed very few particles at the cell surface but a considerable number of cells display virus-like or sub-viral-like particles in intracellular vacuoles.

7) Figure 2B, C, E, 7C, and 8: These panels, which show the levels of released p24, should be accompanied by immunoblotting of cell lysates. This would confirm that Gag expression levels in cells are comparable under the different conditions tested.

A majority of the Figures identified by the reviewers involve HIV-1 DNA transfection with the exception of Figure 2E, which was an infection experiment, not transfection. In the case of infection, we have ensured equal input by using equal MOIs. For all transfections, we have ensured that the transfection efficiencies are equivalent via green fluorescence protein detection after transfecting peGFP n1. We have done immunoblotting to show that the Gag levels in cell lysates are equivalent between the various conditions for Figure 2B, 2C and 7C. Figure Supplement 5 includes analysis for Figure 2B and 2C (now Figures 3B, 3C) and Figure Supplement 7 includes the analysis for Figure 7C (now Figure 11).

8) The discussion should be strengthened in several respects. A) Considering that the main novelty/significance of this study comes from the finding that CCL2 triggers ALIX dissociation from actin cytoskeleton, additional mechanistic explanation (even if speculative) is desirable. In this regard, the discussion on RhoA is probably irrelevant, since RhoA regulates actin cytoskeleton assembly, whereas CCL2 does not seem to affect actin cytoskeleton formation (based on the amount of actin associated with insoluble pellets (Figure 3) or the abundance of stress fibers (Figure 4)), but rather the release of ALIX from actin. B) The authors state that the finding of CCL2-triggered ALIX redistribution has wide-ranging implications in cell and cancer biology but specific discussion on such implications is not provided.

We agree with the reviewer that the discussion on RhoA may be less relevant due to the actin phenotype observed. We thank the reviewer as this comment sheds light on the complexity of the various signal transduction pathways involved. There are numerous pathways that regulate actin cytoskeleton assembly/disassembly. Similarly, there are multiple signal transduction pathways downstream of CCL2 – thus making it difficult to speculate. We have included a brief discussion and have expanded our discussion of the overall implications for cell and cancer biology in the Discussion section.